# Transcriptomics-Morphology Generation Via Treatment Conditioning With Rectified Flow

## Abstract

Predicting cellular responses to drug perturbations requires capturing complex dependencies between transcriptomic and morphological changes that single-modality approaches cannot adequately model. We introduce **PertFlow**, the first unified framework that jointly predicts gene expression profiles and generates cellular morphology images in response to drug treatments, conditioned on control cellular states. Our method integrates control transcriptomic and imaging data through multi-head cross-modal attention mechanisms, learning a shared latent representation that incorporates drug compound features, background cellular profiles, and treatment specifications. From this unified representation, PertFlow employs a regression head for RNA-seq prediction and rectified flow dynamics for stable morphological image generation, with cross-modal consistency losses ensuring coherent molecular and phenotypic predictions. PertFlow enables accurate predictions from either complete multi-modal inputs or single-modality data alone, demonstrating robust cross-modal learning. Our evaluation on paired RNA-seq and Cell Painting fluorescent imaging datasets demonstrates that PertFlow achieves stronger cross-modal consistency and accurate prediction of drug-induced changes compared to diffusion baselines.

## 1 Introduction

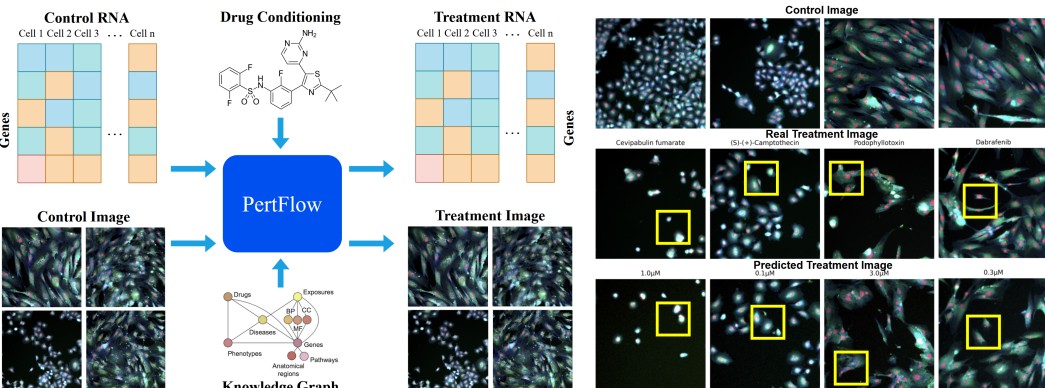

Figure 1: (LEFT) Cross-modal mapping from control RNA-seq and image to treatment RNA-seq and image with drug conditioning through PertFlow. (RIGHT) Comparison of generated treatment vs real treatment images with drug name and concentration. Yellow boxes indicate similar features.

Understanding how drugs alter cellular states is essential for drug discovery, mechanistic understanding, and personalized medicine. Traditional drug response models typically focus on either transcriptomic data or imaging, missing the complex interdependencies between molecular and morphological changes that occur simultaneously in cells. Recent advances in high-throughput profiling now allow paired RNA sequencing and imaging, offering complementary insights: transcriptomics captures molecular mechanisms and gene regulation, while morphology reflects structural and phenotypic changes. These modalities are linked as gene expression can drive morphological transformations, and structural changes can modulate gene activity yet most models treat them in isolation.

Existing methods fall short as transcriptomics-based approaches cannot model morphological effects; image-based models lack molecular interpretability; and cross-modal predictors generate only one modality from another, without joint modeling. Moreover, most studies prioritize genetic over chemical perturbations and analyze rather than predict multi-modal responses. Joint generation of multi-modal responses poses three main challenges: (1) aligning transcriptomic and morphological data across fundamentally different representational spaces; (2) capturing complex drug conditioning involving compound, dose, cell type, and timepoint; and (3) simultaneously predicting discrete gene expression and continuous image data with biological realism and cross-modal consistency.

We introduce **PertFlow** (Figure 1), a novel unified generative framework for jointly predicting treatment gene expression and synthesizing cellular morphology from control conditions, conditioned on drug metadata. Our contributions are: (1) First method to jointly predict transcriptomic and generate morphological responses to chemical perturbations. (2) A shared embedding space integrating control RNA-seq, control images, and drug metadata to model complex dependencies. (3) Multi-token cross-attention to align molecular and morphological features across modalities. We set the benchmark for state-of-the-art performance on the GDPx3 dataset, improving cross-modal alignment and prediction quality over single-modality and diffusion baselines. PertFlow could support downstream applications in virtual drug screening, mechanism discovery, and integrated pharmacological modeling by enabling joint prediction of RNA-seq and image responses to perturbations; a capability, to our knowledge, the first and unique among current methods.

## 2 RELATED WORKS

**Drug-Conditioned and Cross-Modal Modeling:** Recent methods predict transcriptional responses to chemical perturbations but remain largely transcriptomics-focused. Foundational models like scGen (Lotfollahi et al., 2019) pioneered the use of variational autoencoders (VAEs) and latent space vector arithmetic to predict responses to unseen perturbation combinations. Building on this, the Compositional Perturbation Autoencoder (CPA) (Lotfollahi et al., 2021) advanced this concept with a deep generative model to predict single-cell responses to unseen combinations of seen drugs. Its successor, chemCPA (Hetzel et al., 2022), further integrated chemical structures to predict effects for completely unseen drugs. While focused on genetic perturbations, GEARS (Roohani et al., 2024) is another key method using geometric deep learning on gene-gene interaction graphs to predict outcomes for unseen gene perturbations. More recently, PRnet (Qi et al., 2024) employs a perturbation-conditioned generative model to predict expression changes for novel compounds at bulk and single-cell levels. Finally, TranSiGen (Tong et al., 2024) uses self-supervised learning to reconstruct drug-induced profiles from basal expression and compound structure, though limited to denoising and reconstruction. Though Ahlmann-Eltze et al. (2025) demonstrated that performance of deep learning-based models do not significantly yield traditional baselines or statistical methods, the evaluations were predominantly carried on genetic perturbation experiments and leaves chemical perturbation effect untouched.

In parallel, integration of transcriptomic and imaging modalities has emphasized prediction over generation. BLEEP (Xie et al., 2023) applies bi-modal contrastive learning to predict spatial gene expression from H&E images, while SCHAF (Comiter, 2024) is among the few generative models, using GANs to synthesize spatially resolved single-cell omics from histology. TransformerST (Zhao et al., 2024) fuses histology with gene expression for super-resolution predictions, prioritizing data enhancement. Multi-modal perturbation frameworks such as Perturb-multi-modal (Saunders et al., 2025) and CRISPR ST (Binan et al., 2025) integrate imaging and RNA-seq to study genetic perturbations, but focus on measurement rather than synthesis and mainly on genetic rather than chemical interventions. Fusion-based methods (Lu et al., 2024) combine chemical, transcriptomic, and other biological data for prediction and classification, yet cross-modal generative modeling of cellular responses remains unaddressed.

**Morphological Profiling and Generative Frameworks:** Cellular imaging provides critical insights into drug mechanisms, with Cell Painting (Bray et al., 2016) capturing multiplexed phenotypes under perturbations and widely used in virtual screening. Advances in deep learning have enhanced morphological profiling through convolutional models and computer vision (Tang et al., 2024), where tools such as CellProfiler (McQuin et al., 2018) automate analysis and Cellpose (Stringer et al., 2021) improves segmentation. However, generative modeling of cellular images conditioned on

perturbations remains limited. For instance, recent efforts focus purely on image-to-image translation, such as PhenDiff (Bourou et al., 2023), which uses a conditional diffusion model, and Lamiable et al. (2023), which employs conditional GANs. Critically, these methods generate a cell image in one condition given an image from another but operate without any transcriptomic context.

Progress in generative frameworks highlights potential for this gap: diffusion models achieve state-of-the-art performance in image, protein, and molecule generation (Guo et al., 2024), but suffer from high computational cost and slow sampling. Rectified Flow (Liu et al., 2022) offers a more efficient alternative by learning straight-line transport between distributions, reducing sampling steps without loss of quality. This efficiency stems from flow matching (Lipman et al., 2022), which linearly interpolates between noise and data, making rectified flow especially suited for large-scale drug screening. While most current methods are single-modal, advances such as Stable Diffusion 3 (Esser et al., 2024) demonstrate the feasibility of multi-modal generative modeling, opening opportunities for predictive simulation of cellular responses.

## 3 METHODS

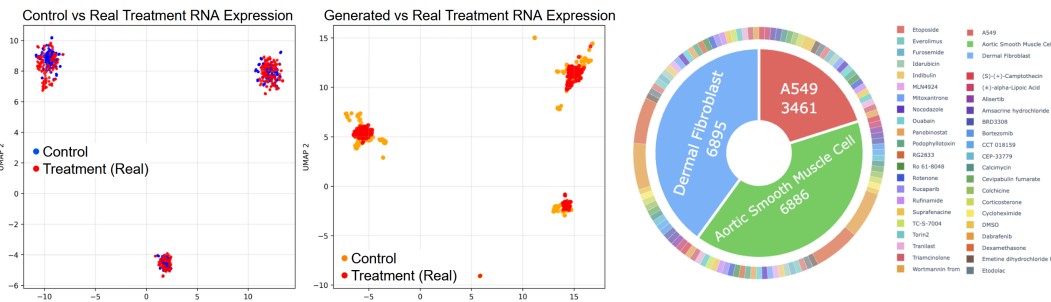

Figure 2: (LEFT) UMAP representation of control vs real treatment vs generated treatment gene expression data. (RIGHT) Distribution of cell lines and compounds in dataset.

**Problem Formulation.** We formalize the drug conditioned multi modal generation problem as learning a mapping from control cellular states to treatment responses across transcriptomic and morphological modalities. Given control gene expression $\mathbf{x}_{\text{rna}}^{\text{ctrl}} \in \mathbf{R}^G$ where $G$ is the number of genes, control cellular images $\mathbf{x}_{\text{img}}^{\text{ctrl}} \in \mathbf{R}^{C \times H \times W}$ with $C$ channels and spatial dimensions $H \times W$, and drug conditioning information $\mathbf{c} = \{c_{\text{compound}}, c_{\text{cell}}, c_{\text{conc}}, c_{\text{time}}\}$ including compound identity, cell line, concentration, and timepoint, our objective is to generate treatment outcomes, where $f_\theta$ represents our unified generative model parameterized by $\theta$: $\mathbf{x}_{\text{rna}}^{\text{treat}}, \mathbf{x}_{\text{img}}^{\text{treat}} = f_\theta(\mathbf{x}_{\text{rna}}^{\text{ctrl}}, \mathbf{x}_{\text{img}}^{\text{ctrl}}, \mathbf{c})$

**Dataset Description.** Our study leverages the Ginkgo Data Platform (GDP) series (Model & Biologics, 2025), a multimodal dataset integrating transcriptomic profiles (GDPx1/GDPx2) and four-channel fluorescence microscopy images (GDPx3) from drug-treated cell cultures. We curated paired bulk RNA-seq and Cell Painting imaging dataset from 3 cell lines and 40 drugs as illutrated in Figure 2 (RIGHT). We implemented cross-modal pairing by identifying overlapping compounds and experimental conditions, standardizing metadata (concentration units, cell line nomenclature, temporal alignment), and establishing DMSO controls as baseline references. Transcriptomic pre-processing follows established protocols like total count normalization to $10^6$ reads per sample, log1p transformation, and highly variable gene selection (n=8000) using scanpy, to focus on the most informative genomic features. Image preprocessing addresses 16-bit microscopy data through proper intensity scaling (16-bit to [-1,1] range), percentile-based contrast enhancement ($1^{st}$-$99^{th}$ percentile) applied per channel, and bilinear interpolation to uniform spatial dimensions. The dataset consists of 17242 paired samples that were split by 80:20 for training and testing.

Drug compounds are represented through multi-modal molecular encodings combining structural and physicochemical information. We extract Morgan and RDKit molecular fingerprints (1024 bits each) from canonical SMILES strings, providing binary structural descriptors capturing substructural patterns and pharmacophoric features. Molecular descriptors include eighteen 2D properties (molecular weight, logP, topological polar surface area, hydrogen bond donors/acceptors, rotatable bonds, aromatic rings, and complexity measures) and five 3D properties when available from SDF

structures. For compounds lacking preprocessed molecular data, we implement on-demand SMILES processing with RDKit to ensure comprehensive coverage. Missing molecular information is handled through zero-padding with appropriate masking, while molecular descriptors are normalized using dataset-wide statistics to ensure stable training dynamics across diverse chemical spaces. The dataset allows stratified paired control-treatment comparisons, where drug-treated samples are systematically matched with vehicle DMSO controls from identical cell lines and experimental conditions. This preserves the combinatorial structure of cell line-compound-dose-time relationships across training and validation partitions, ensuring robust model generalization across the full experimental parameter space.

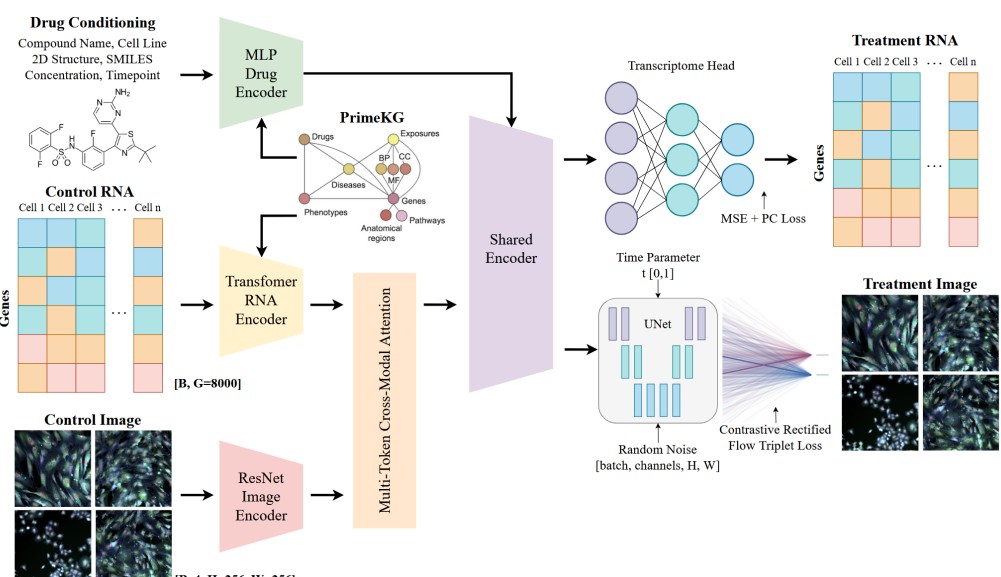

Figure 3: PertFlow architecture for drug conditioned control RNA-image to treatment RNA-image. Input RNA-seq and image going through their respective encoders; output from the two encoders then pass through the multi-token cross-modal attention, before entering the shared encoder along with conditioning information which passes through the drug encoder. The transcriptome head uses MSE loss to predict treatment RNA, and the image UNet (Huang et al., 2020) with noise and time parameter input uses triplet contrastive loss and rectified flow dynamics to generate treatment cellular image, from the shared embeddings, respectively. More details in the appendix.

**Architecture.** PertFlow (Figure 3) uses a shared representation learning paradigm with 3 components: (1) individual modality encoders that process control RNA-seq and imaging data, (2) cross-modal attention mechanism that aligns features across modalities, (3) generation heads that produce treatment RNA-seq via direct prediction and treatment images via rectified flow dynamics.

PertFlow processes control RNA-seq and cellular images through specialized encoders. The RNA-seq encoder applies multi-layer self-attention to capture gene-gene interactions, embedding each gene expression value into high-dimensional space and processing through attention and feedforward layers with residual connections. Attention-weighted pooling aggregates the final embeddings into a single RNA-seq representation encoding the transcriptomic state. Control images are encoded through a ResNet-style convolutional architecture with global pooling, extracting hierarchical visual features from cellular morphology. Both modalities are enhanced through PrimeKG knowledge graph integration. For RNA-seq, a heterogeneous graph neural network processes protein-protein interactions and pathway information. For drugs, the graph encoder processes drug-protein interactions and pharmacological relationships. Knowledge embeddings are integrated additively with learned representations using weighting factors $\alpha_{\text{drug}} = 0.3$ and $\alpha_{\text{RNA}} = 0.3$. Drug conditions are encoded through a fusion module combining learned embeddings for compound and cell line identities with encoded concentration and time parameters.

Each modality embedding is projected to $K$ token representations to prevent information bottlenecks. RNA and image tokens undergo self-attention within modalities, then bidirectional cross-

attention: RNA tokens attend to image tokens and vice versa, integrating information across modalities. Cross-attended tokens combine with original tokens through residual connections, then aggregate to single vectors via attention pooling. The enhanced RNA and image embeddings concatenate with drug conditioning and process through a shared encoder—a multi-layer perceptron producing unified representation $\mathbf{h}_{\text{shared}}$ where all modalities converge. This representation branches to two task-specific generation heads. Treatment transcriptomes are generated through direct regression from $\mathbf{h}_{\text{shared}}$ via a fully-connected prediction head. For images, PertFlow adapts rectified flow, defining linear interpolation paths $\mathbf{x}_t = (1 - t)\mathbf{z}_0 + t\mathbf{x}_{\text{img}}^{\text{treat}}$ for $t \in [0, 1]$ with constant velocity $\mathbf{v}_t = \mathbf{x}_{\text{img}}^{\text{treat}} - \mathbf{z}_0$. A UNet predicts this velocity field conditioned on noisy state $\mathbf{x}_t$, timestep $t$, and $\mathbf{h}_{\text{shared}}$ injected through cross-attention layers. During inference, an adaptive DOPRI5 ODE solver integrates the learned velocity field from noise to treatment image with automatic step size adjustment.

Transcriptome prediction combines MSE with Pearson correlation loss:

$$\mathcal{L}_{\text{rna}} = 0.9 \cdot \text{MSE}(\mathbf{x}_{\text{rna}}^{\text{treat}}, \hat{\mathbf{x}}_{\text{rna}}^{\text{treat}}) + 0.1 \cdot (1 - \text{PC}(\mathbf{x}_{\text{rna}}^{\text{treat}}, \hat{\mathbf{x}}_{\text{rna}}^{\text{treat}})) \tag{1}$$

where MSE ensures gene-level accuracy and correlation preserves relative expression patterns. The rectified flow objective trains velocity prediction:

$$\mathcal{L}_{\text{img}} = \mathbb{E}_{t, \mathbf{z}_0, \mathbf{x}_{\text{img}}^{\text{treat}}} \left[ \|\mathbf{v}_\theta(\mathbf{x}_t, t, \mathbf{h}_{\text{shared}}) - (\mathbf{x}_{\text{img}}^{\text{treat}} - \mathbf{z}_0)\|^2 \right] \tag{2}$$

over random timesteps $t \sim \text{Uniform}(0, 1)$ and noise samples $\mathbf{z}_0$.

Triplet contrastive consistency enforces coherent multi-modal predictions by comparing aligned versus misaligned features:

$$\mathcal{L}_{\text{triplet}} = \mathbb{E}[\max(0, \text{margin} - (\mathcal{L}_{\text{neg}} - \mathcal{L}_{\text{pos}}))] \tag{3}$$

where $\mathcal{L}_{\text{pos}}$ is prediction error with aligned RNA-image features and $\mathcal{L}_{\text{neg}}$ with shuffled features. The complete objective combines all losses:

$$\mathcal{L}_{\text{total}} = w_{\text{rna}}\mathcal{L}_{\text{rna}} + w_{\text{img}}\mathcal{L}_{\text{img}} + w_{\text{triplet}}\mathcal{L}_{\text{triplet}} \tag{4}$$

with weights $w_{\text{rna}} = 0.5$, $w_{\text{img}} = 0.5$, $w_{\text{triplet}} = 0.05$. All parameters are jointly optimized for end-to-end multi-modal learning.

**Training Parameters.** We use 4 attention heads with embedding dimension 128, applying 1 layer of self-attention followed by attention-based pooling. Multi-token representations use $K = 16$ tokens with hidden dimension 256 and 8 attention heads. The rectified flow UNet uses 192 base channels with channel multipliers $(1, 2, 2, 2)$, attention at 16×16 resolution, and cross-attention conditioning at layers 2, 3, 4, and 5. Models are trained with AdamW optimizer ($\beta_1 = 0.9$, $\beta_2 = 0.95$), learning rate $10^{-4}$ with cosine annealing, and automatic mixed precision. Cross-modal consistency weights are gradually increased during training to ensure stable convergence. RNA-seq generation requires a single forward pass, while image generation uses 7-10 DOPRI5 steps with relative tolerance $10^{-3}$ and absolute tolerance $10^{-4}$ for high-quality synthesis. The models were trained with an effective batch size of 32 taking 5 hours on 8 H100 NVIDIA GPUs with 80GB VRAM.

# 4 EXPERIMENTS

We emphasize, since our method is the first to introduce the problem of multi-modal RNA-Image generation with drug conditioning, we have **no previous multi-modal method** to fairly compare to as baseline. To create baselines we trained three diffusion models, along with ablations of knowledge graph module, triplet contrastive objective, and Pearson correlation loss, with RNA only and Image only models. We still included PRNet Qi et al. (2024) and PhenDiff Bourou et al. (2023) as unimodal method for reference. We set the state-of-the-art performance for this problem on the GDPx3 dataset.

**Drug effects on gene expression and cell morphology:** We trained PertFlow (control RNA-seq and image to treatment RNA-seq and image), PertRNA (control RNA-seq to treatment RNA-seq), PertImage (control image to treatment image), and their respective ablations omitting the knowledge-graph and contrastive rectified flow objective. PertFlow demonstrates strong performance in predicting treatment gene expression from control conditions in Table 1. Achieving Pearson correlation

Table 1: PertFlow & PertRNA metrics (mean ± std)

| Model | MSE ↓ | RMSE ↓ | MAE ↓ | Pearson $r$ ↑ | Spearman $r$ ↑ |
|---|---|---|---|---|---|
| MLP Baseline | 63.846 ± 0.656 | 8.513 ± 0.660 | 5.281 ± 0.665 | 0.224 ± 0.040 | 0.508 ± 0.054 |
| VAE PRNet | 25.655 ± 0.453 | 5.075 ± 0.597 | 3.244 ± 0.977 | 0.452 ± 0.034 | 0.697 ± 0.049 |
| PertRNA(-PC) | 0.412 ± 0.525 | 0.598 ± 0.912 | 0.286 ± 0.374 | 0.511 ± 0.144 | 0.502 ± 0.097 |
| PertRNA(-KG) | 0.360 ± 1.274 | 0.475 ± 0.366 | 0.112 ± 0.106 | 0.770 ± 0.098 | 0.791 ± 0.063 |
| PertRNA | 0.311 ± 0.956 | 0.472 ± 0.271 | 0.111 ± 0.025 | 0.779 ± 0.081 | **0.795 ± 0.026** |
| PertFlow(-KG) | 0.262 ± 0.394 | 0.470 ± 0.202 | 0.114 ± 0.034 | 0.772 ± 0.107 | 0.735 ± 0.066 |
| PertFlow | **0.231 ± 0.708** | **0.462 ± 0.107** | **0.110 ± 0.166** | **0.780 ± 0.264** | 0.792 ± 0.041 |

Table 2: PertFlow & PertImage metrics (mean ± std)

| Model | SSIM ↑ | PSNR ↑ | LPIPS → | FID ↓ |
|---|---|---|---|---|
| UNet Baseline | 0.085 ± 0.013 | 03.55 ± 0.73 | 2.125 ± 0.158 | 583.21 |
| UNet PhenDiff | 0.189 ± 0.025 | 9.64 ± 0.64 | 0.613 ± 0.531 | 62.50 |
| PertDiff$_N$ | 0.010 ± 0.003 | 06.62 ± 0.73 | 1.087 ± 0.096 | 246.01 |
| PertDiff$_{x0}$ | 0.192 ± 0.075 | 10.71 ± 0.54 | 0.505 ± 0.045 | 73.63 |
| PertDiff$_V$ | 0.194 ± 0.071 | 11.33 ± 0.42 | 0.499 ± 0.040 | 55.92 |
| PertImage(-triplet) | 0.120 ± 0.102 | 08.22 ± 0.13 | 0.308 ± 0.178 | 106.72 |
| PertImage(-KG) | 0.182 ± 0.099 | 11.05 ± 0.58 | 0.498 ± 0.035 | 50.38 |
| PertImage | 0.187 ± 0.095 | 11.46 ± 0.61 | 0.509 ± 0.043 | 46.88 |
| PertFlow(-KG) | **0.206 ± 0.094** | 11.51 ± 0.70 | 0.505 ± 0.043 | 31.59 |
| PertFlow | 0.205 ± 0.097 | **11.66 ± 0.68** | **0.511 ± 0.038** | **24.06** |

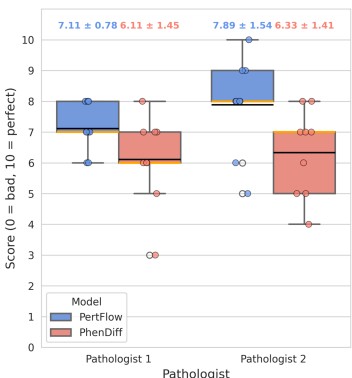

Figure 4: Pathologist image similarity score of generated vs ground truth cellular morphology after treatment. (Mean solid lines, Median orange line)

Table 3: Effect of Loss Weight Ratios

| $w_{rna}$ | $w_{img}$ | MSE ↓ | Pearson r ↑ | Spearman r ↑ | SSIM ↑ | PSNR ↑ | FID ↓ |
|---|---|---|---|---|---|---|---|
| 0.4 | 0.6 | 0.268±0.752 | 0.761±0.175 | 0.773±0.047 | **0.215±0.094** | **11.94±0.65** | **21.73** |
| 0.5 | 0.5 | 0.231±0.708 | 0.780±0.264 | 0.792±0.041 | 0.205±0.097 | 11.66±0.68 | 24.06 |
| 0.6 | 0.4 | **0.219±0.694** | **0.791±0.251** | **0.801±0.038** | 0.193±0.101 | 11.29±0.73 | 27.15 |

Table 4: Effect of Knowledge Graph Embedding Weight

| $\alpha_{drug}$ | MSE ↓ | Pearson r ↑ | Spearman r ↑ | SSIM ↑ | PSNR ↑ | FID ↓ |
|---|---|---|---|---|---|---|
| 0.1 | 0.289±0.784 | 0.741±0.287 | 0.753±0.052 | 0.187±0.103 | 11.12±0.76 | 29.31 |
| **0.3** | **0.231±0.708** | **0.780±0.264** | **0.792±0.041** | **0.205±0.097** | **11.66±0.68** | **24.06** |
| 0.5 | 0.248±0.761 | 0.759±0.279 | 0.769±0.048 | 0.194±0.101 | 11.38±0.73 | 28.47 |

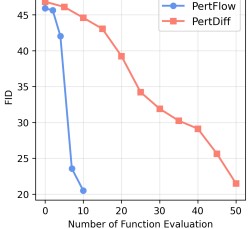

Figure 5: FID vs NFE

(0.780 ± 0.264) and Spearman correlation (0.792 ± 0.041) across drug perturbations, measuring with all genes. MSE (0.231 ± 0.708) and MAE (0.110 ± 0.166) indicate robust prediction accuracy for transcriptomic responses. While the PertRNA baseline achieved correlation metrics (Pearson r (0.779 ± 0.081), Spearman r (0.795 ± 0.026)), PertFlow's joint modeling approach maintains competitive performance while simultaneously generating cellular morphological responses compared to baselines and ablations.

PRNet Qi et al. (2024) is a perturbation-conditioned generative model comprising three components: a Perturb-adapter encoding compound SMILES structures to latent embeddings, a Perturb-encoder mapping perturbation effects to latent space, and a Perturb-decoder estimating Gaussian distributions of perturbed transcriptional profiles. The simple MLP baseline replaces the full encoder-decoder architecture with a multilayer perceptron that directly learns perturbation effects on gene expression using MSE loss. This simplified architecture achieved only Pearson correlation of 0.224±0.040 and Spearman correlation of 0.508±0.054, with MSE of 63.846±0.656. The VAE PRNet baseline implements the complete VAE-inspired framework with encoder-decoder architecture estimating Gaussian distributions parameterized by mean and variance, but operates on transcriptional data alone without multi-modal integration. VAE PRNet substantially improved over the MLP variant with Pearson correlation of 0.452±0.034 and Spearman correlation of 0.697±0.049 (MSE 25.655±0.453), yet remained far below PertRNA (Pearson 0.779±0.081, Spearman 0.795±0.026) and PertFlow (Pearson 0.780±0.264, Spearman 0.792±0.041). Both PRNet variants predict treatment RNA profiles but lack the cross-modal enhancement, knowledge graph integration, and shared representation learning that enables PertFlow to achieve coherent simultaneous prediction of transcriptional and morphological responses while maintaining competitive performance on both modalities.

PertFlow also outperformed the PertImage baseline in the image generation task achieving higher SSIM (0.205 ± 0.097), PSNR (11.66 ± 0.68), LPIPS (0.511 ± 0.038), and lower FID (24.06). This demonstrates that incorporating cross-modal information and drug conditioning does not compromise transcriptomic prediction and cellular image generation quality, while enabling the unique

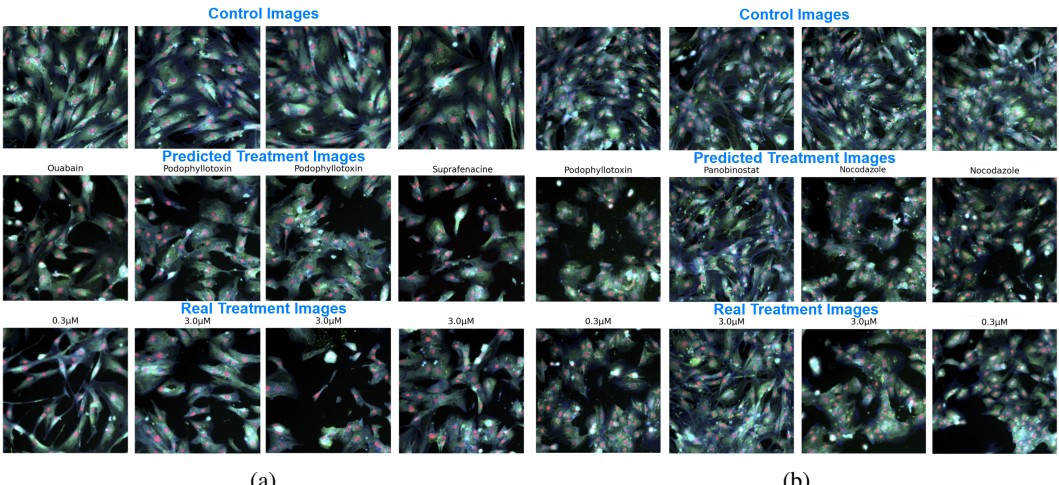

Figure 6: Generated vs real treatment for (a) Aortic Smooth Muscle Cell (b) Dermal Fibroblasts.

capability of joint multi-modal generation. The strong correlation values and lower FID indicates that PertFlow successfully captures the complex cellular relationships between drug perturbations and gene expression changes.

We compare our PertFlow model with three PertDiff diffusion variants in Table 2. Diffusion serves as the baseline against our contrastive rectified flow method. The standard DDPM formulation trains the model to predict the injected noise $\epsilon \sim N(0, I)$ from the noisy input $x_t$. This requires disentangling signal from noise across noise levels, causing instabilities when signal-to-noise ratios are low. The weak performance of PertDiff$_N$ (SSIM: 0.010, FID: 246.01) illustrates these challenges, as the model fails to generate coherent cellular structures from pure noise predictions. The direct $x_0$ parameterization predicts the clean target $x_0$ from noisy $x_t$. While more stable, it forces the model to implicitly learn the full denoising trajectory. PertDiff$_{x0}$ achieves better but still limited results (SSIM: 0.192, FID: 73.63), reflecting the lack of strong theoretical grounding. The velocity ($v$) parameterization improves training stability by predicting $v = \alpha_t \epsilon - \sigma_t x_0$, which balances objectives across timesteps, reduces variance, and improves gradient flow. PertDiff$_V$ shows marked improvement (SSIM: 0.194, FID: 55.92), validating this formulation for biological image generation. Compared to baselines and ablations we observe that PertFlow successfully learns more meaningful representations that generalize across different compounds, concentrations, cell lines, and timepoints in the dataset.

PhenDiff Bourou et al. (2023) is a conditional diffusion model that performs image-to-image translation to identify phenotypic shifts in microscopy images. The model operates through two stages: an inversion phase that maps real source images to Gaussian latent representations using DDIM deterministic sampling, followed by a generation phase that synthesizes images in the target condition. The UNet-based PhenDiff baseline predicts treatment-induced morphological changes without incorporating transcriptional information, serving as a purely vision-based approach. When evaluated on the GDPx3 dataset, UNet PhenDiff achieved moderate performance with SSIM of 0.189±0.025, PSNR of 9.64±0.64, and FID of 62.50, substantially outperforming the vanilla UNet baseline (SSIM 0.085±0.013, FID 583.21) but lagging behind multi-modal approaches. The deterministic UNet baseline performed even worse since it is trained with a simple MSE loss.

Table 3 shows equal loss weighting (0.5–0.5) achieves the best trade-off. Higher $w_{\text{rna}}$ improves RNA accuracy but worsens image quality (higher FID). Higher $w_{\text{img}}$ improves SSIM/PSNR but reduces RNA correlation metrics. Table 4 shows the effect of KG weight $\alpha_{\text{drug}} = 0.3$ is optimal. Smaller values (0.1, 0.2) underuse the biological prior, while larger values (0.4, 0.5) introduce noise that harms both modalities.

Figure 4 shows similarity rating by ACVP board certified pathologists in blind review (10-point scale of similarity regarding morphology, detail and plausibility, with respect to ground truth Cell Painting images under corresponding chemical perturbations; 0 indicates poorest and 10 indicates the best). Two pathologists reported median score of 7.11 and 7.89 for PertFlow generated images, outperforming the baseline method PhenDiff (Bourou et al., 2023) (median scores of 6.11 and 6.33), which confirmed the overall satisfying quality of cellular morphology images generated by PertFlow. Step-wise comparison (Figure 5) highlights the difference in inference dynamics. PertFlow generates recognizable structures by NFE 10 showing nuclear boundaries and cytoplasmic organization with near-final morphology. Pert-

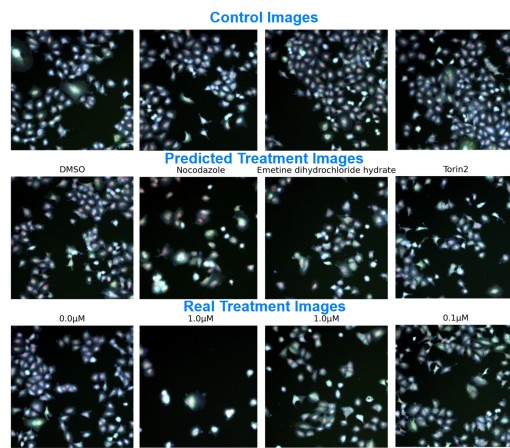

Figure 7: Generated vs real treatment for A549.

Diff requires more steps to reach similar organization. Both methods preserve multi-channel fluorescence distributions.

**Recovering drug-induced phenotype and morphology:** Figure 1 shows the comparison of generated treatment vs real treatment images with drug name and concentration. Yellow boxes indicate similar cellular features due to drug perturbations in real and generated images. From left to right the drugs have the following effect on the cellular morphology: (1) Cevipabulin is a microtubule-destabilizing agent that binds to tubulin (Yang et al., 2021), disrupting microtubule dynamics, which leads to mitotic arrest and apoptosis in cancer cells. It shows anti-proliferative effects by inhibiting microtubule polymerization. (2) S-Camptothecin and its stereoisomers inhibit DNA topoisomerase (Hansch & Verma, 2007), causing DNA damage during replication. This leads to DNA double-strand breaks, S-phase cell cycle arrest, and apoptosis, especially in rapidly dividing cells. (3) Podophyllotoxin binds to tubulin and inhibits microtubule assembly (Desbene & Giorgi-Renault, 2002), resulting in mitotic arrest at metaphase and subsequent apoptosis. It serves as a precursor for etoposide, a topoisomerase II inhibitor. (4) Dabrafenib selectively inhibits mutant BRAF kinase (commonly V600E mutation) (Planchard et al., 2022), blocking MAPK/ERK signaling pathway, leading to decreased tumor cell proliferation and inducing apoptosis in BRAF-mutated cancer cells.

The ACVP certified pathologists provided the following descriptions of generated and real treatment image results in Figures 6 and 7. In Figure 6a control images exhibited typical fusiform cells with parallel alignment and organized architecture. Both predicted and real Ouabain (0.3 $\mu$M) treatment showed decreased cellular density, reduced cell size, increased intercellular spacing, nuclear condensation, and disrupted parallel orientation with cells. Similarly, predicted and actual Podophyllotoxin (3.0 $\mu$M) treatment displayed increased intercellular spacing, elevated multi-nucleated cell populations indicative of cellular injury, loss of fusiform morphology, and decreased cellular density. Suprafenacine (3.0 $\mu$M) predictions and actual treatment both revealed cellular disorganization, loss of fusiform shape, cellular fragmentation, nuclear size reduction, and decreased cell-to-cell contact.

In Figure 6b control images displayed characteristic fibroblast morphology with fusiform cells, organized cellular arrangement, and appropriate cell-to-cell contact. Both predicted and real Podophyllotoxin (0.3 $\mu$M) treatment exhibited decreased cellular density, reduced cell-to-cell contact, and loss of fusiform morphology while maintaining nuclear size. Panobinostat (3.0 $\mu$M) predictions and actual treatment showed minimal morphological deviation from control despite the higher dosage, with the model correctly preserving the relatively unaltered cellular architecture. Nocodazole treatment at 3.0 $\mu$M demonstrated strong concordance between predicted and real images, both displaying substantial loss of cellular density and fusiform morphology, with cells adopting rounded, blob-like shapes while nuclear size remained similar. At the lower nocodazole dose (0.3 $\mu$M), both predicted and actual treatments showed attenuated phenotypic changes including decreased cellular density, partial retention of fusiform morphology, reduced cellular elongation, and less pronounced blob-like transformation compared to the higher dose.

In Figure 7 control images exhibited typical A549 morphology with appropriate cellular density and organization. DMSO treatment (0.0 $\mu$M) served as vehicle control, with both predicted and real

images showing no morphological deviation from control conditions. Nocodazole (1.0 $\mu$M) predictions and actual treatment both displayed preserved cellular morphology with substantially decreased cellular density and reduced cellularity. Emetine dihydrochloride hydrate (1.0 $\mu$M) showed concordance between predicted and real images, both exhibiting decreased cellular density and diminished cellular cohesiveness. Torin2 (0.1 $\mu$M) predictions and actual treatment demonstrated increased angular cytoplasmic projections, moderate reduction in cellular density, and decreased cellular cohesion. The morphological agreement between predicted and real treatment conditions across varying drug concentrations validates the model's dose-dependent prediction capability.

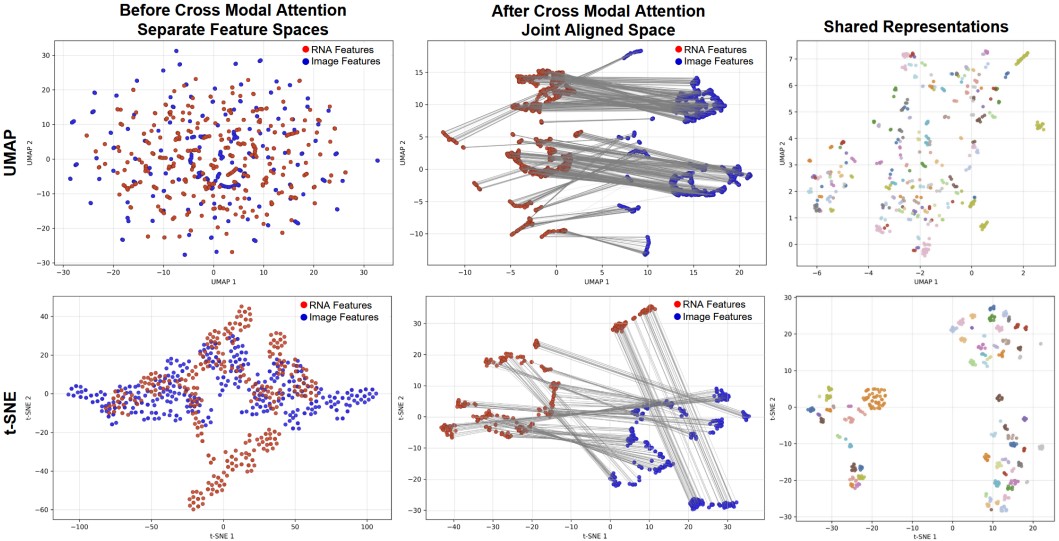

Figure 8: UMAP and t-SNE of RNA and image embeddings before and after cross modal attention

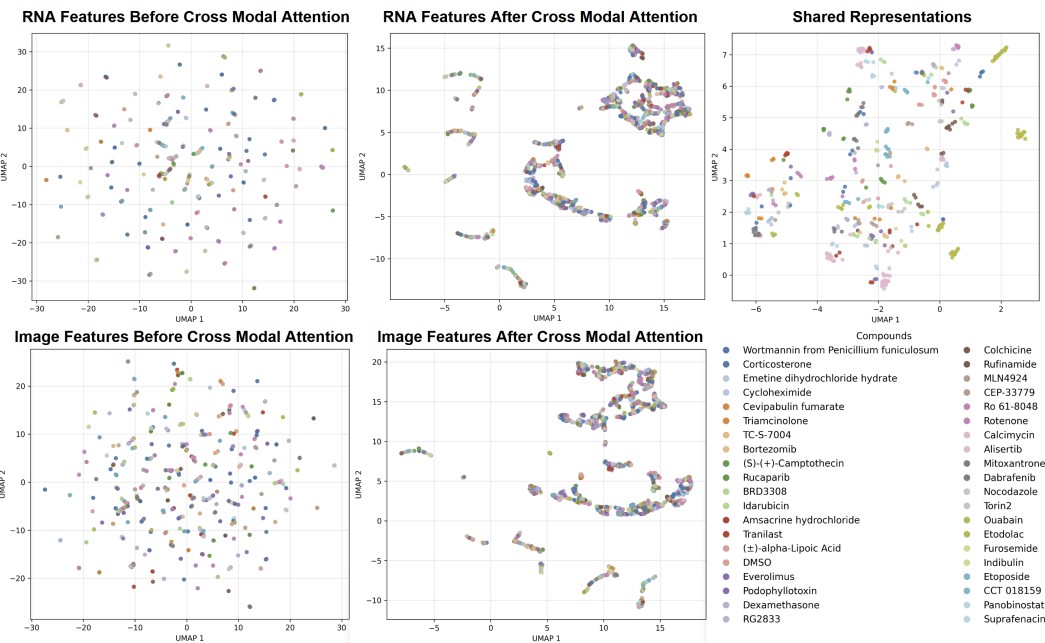

Figure 9: UMAP of RNA and image embeddings before and after cross modal attention block

Figure 2 UMAPs (McInnes et al., 2018) demonstrate PertFlow's biological coherence in generating treatment responses. The left panel shows clear control-treatment cluster separation, indicating distinct transcriptomic signatures captured in embedding space. The right panel shows generated

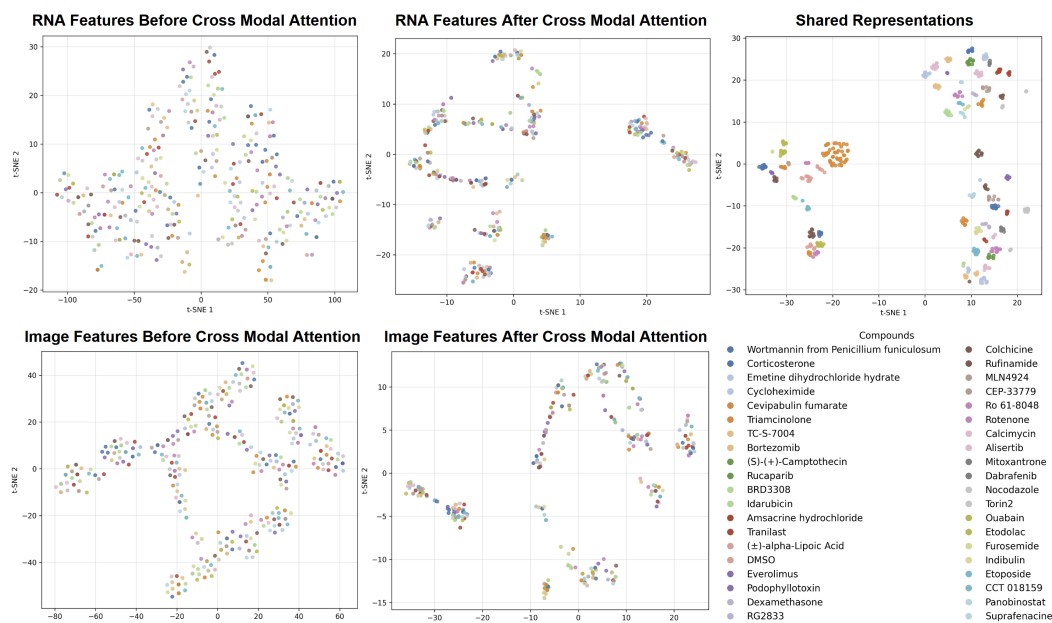

Figure 10: t-SNE of RNA and image embeddings before and after cross modal attention block

treatment samples clustering with real samples, particularly in high-response regions, demonstrating that cross-modal attention effectively leverages imaging to predict directional gene expression changes and produce biologically realistic treatment profiles.

Figures 8, 9, 10 UMAPs and t-SNE illustrate cross-modal attention's effect on feature alignment. The top panels show transformation from separate feature spaces (left) to joint aligned space (middle), with RNA-seq (blue) and image features (red) demonstrating successful correspondence learning. The middle and bottom panels reveal that cross-modal attention transforms scattered, unstructured distributions into highly structured, clustered organizations in both modalities, indicating that attention not only aligns features across modalities but enhances the internal structure and discriminative power of each individual feature space. The shared representation space (right) demonstrates successful integration of RNA, imaging, and drug modalities. Compound-based organization indicates that cross-modal attention effectively combines gene expression, imaging, and drug conditioning into a unified latent space where biologically similar samples cluster together regardless of modality, enabling improved treatment response prediction and generation.

Cross-modal attention shows correspondence between RNA-image modalities, preserving distinct structures. Pairwise alignment distances average 16.48 (UMAP) and 37.44 (t-SNE), with positive modality separation scores (UMAP: 0.81, t-SNE: 0.59) confirming preserved modality-specific features. Compound clustering improves substantially, with silhouette scores rising from -0.74 to -0.20 (UMAP) and -0.81 to -0.28 (t-SNE). The shared space separation ratio of 1.33 validates treatment-specific representations consistent across both modalities. Alignment metrics used standardized embeddings projected into 2D via UMAP and t-SNE. Pairwise alignment distances were averaged Euclidean distances between corresponding RNA-image pairs. Modality separation used silhouette scores with binary labels (0=RNA, 1=Image), where positive values indicate preserved modality-specific structures. Compound clustering quality was assessed via silhouette scores with treatment labels. Shared space separation ratio was calculated as mean inter-compound cosine distance divided by mean intra-compound distance, values >1 indicate successful treatment discrimination.

**Gene Enrichment Analysis.** Gradient-based feature importance analysis (Figure 11) of gene contributions to morphological recovery revealed cell line-specific transcriptional responses to pharmaceutical compounds, where gene importance scores were extracted from latent representations, top 200 genes were selected per drug-cell line pair, and GSEA was performed using MSigDB Hallmark 2020 collection. A549 lung cancer cells showed limited pathway enrichment dominated by TNF-alpha/NF-kB signaling consistent with KRAS-mutant inflammatory dependency, while HASMC and

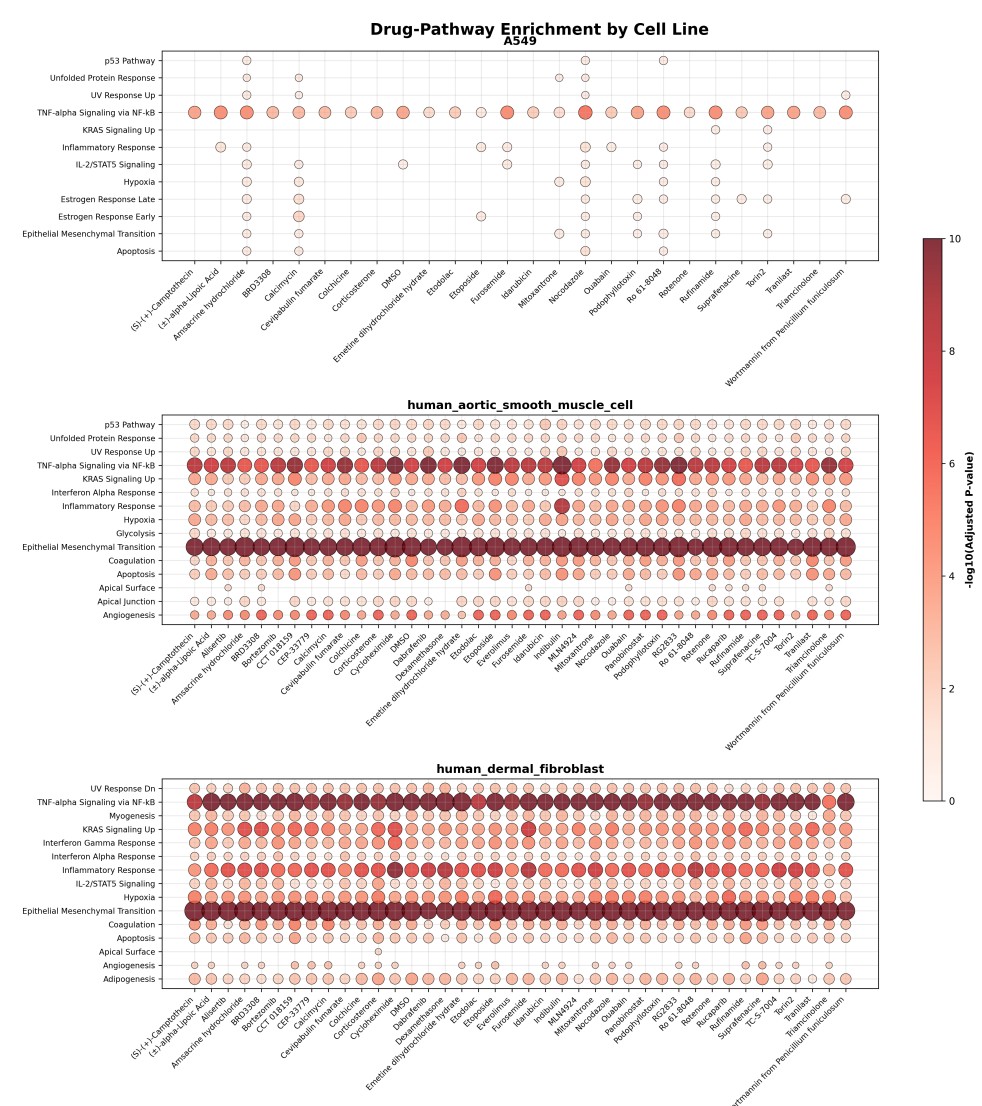

Figure 11: Dotplot for geneset enrichment anlaysis from activated genes during inference

dermal fibroblasts demonstrated robust multi-pathway responses with EMT as the most consistently enriched pathway across treatments, followed by TNF-alpha/NF-kB as the second universal stress signature. Cell-specific responses included vascular pathways (angiogenesis, coagulation, hypoxia) in HASMC and tissue remodeling pathways (myogenesis, adipogenesis, UV response, interferon gamma) in fibroblasts, validating the model's biological accuracy through known drug mechanisms: DNA-damaging agents activated p53/apoptosis pathways in TP53-intact cells, dabrafenib triggered compensatory KRAS signaling, and anti-inflammatory compounds modulated IL-2/STAT5 pathways. Concordance between transcriptional signatures and morphological predictions (cell number decreases with apoptosis-inducing drugs like nocodazole) confirmed that the multimodal architecture captures functionally relevant biological relationships, demonstrating that drug response profiling reveals both universal cellular stress responses and cell type-specific vulnerabilities relevant to therapeutic resistance mechanisms. Further analysis in the appendix.

## 5 DISCUSSION

PertFlow represents a foundational step toward unified modeling of multi-modal cellular drug responses, bridging the molecular (transcriptomic) and phenotypic (morphological) effects of chemical perturbations. Unlike previous approaches that treat these modalities in isolation or only predict one from the other, PertFlow achieves simultaneous, drug-conditioned generation of both gene expression profiles and cellular morphology. The integration of control transcriptomic and imaging data into a shared embedding space, combined with rectified flow dynamics, enables biologically consistent synthesis of treatment outcomes. Despite this progress, several challenges remain. First, generalization to unseen cell lines or novel compounds is limited by the scarcity of paired multimodal datasets with shared metadata. While PertFlow can still infer morphological changes from control data alone, future work should explore more targeted integration of chemogenomic databases (e.g., ChEMBL, DrugBank) that explicitly encode compound-target binding affinities and structure-activity relationships, building upon our current PrimeKG integration which provides broader biological context but lacks fine-grained chemical similarity information critical for generalizing to truly novel compounds. Second, while PertFlow enables modality translation, aligning embeddings across modalities may inadvertently entangle task-relevant factors. Disentangling causal latent factors remains an open question for cross-modal modeling. Third, the in vitro context of our experiments may not capture drug effects requiring complex microenvironmental interactions, such as immune modulation. Extending PertFlow to model cell-cell communication or tissue-level organization could enhance translational utility. Overall, PertFlow sets the stage for future cross-modal generative modeling in drug discovery, offering a unified framework for understanding how molecular mechanisms manifest as observable phenotypes under pharmacological perturbation.

## 6 CONCLUSION

We introduced PertFlow, the first unified generative framework for jointly modeling transcriptomic and morphological drug responses using cross-modal attention and rectified flow dynamics. By aligning control RNA-seq and image features through a shared embedding space conditioned on drug metadata, PertFlow enables simultaneous prediction of treatment gene expression and synthesis of cellular morphology. Extensive evaluation on the GDPx3 dataset demonstrates strong cross-modal consistency, biologically realistic image generation, and competitive transcriptomic prediction performance, outperforming single-modality and diffusion baselines. UMAP analysis of cross-modal attention and shared embeddings further strengthen our hypothesis. Our results highlight PertFlow's potential for virtual drug screening, mechanistic hypothesis generation, and multimodal perturbation analysis, paving the way for more integrative and interpretable approaches to pharmacological modeling.

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

# A APPENDIX

## A.1 ETHICS STATEMENT

We used large language models solely for manuscript proofreading and grammar checking, with no involvement in code or content generation.

## A.2 REPRODUCIBILITY STATEMENT

We will release the code and pretrained model weights upon acceptance.

## A.3 ARCHITECTURE DETAILS

Our RNA-seq encoder processes gene expression data through multi-layer self-attention (Vaswani et al., 2017) to capture gene-gene interactions:

$$\mathbf{E}_{\text{gene}} = \text{GeneEmbedding}(\mathbf{x}_{\text{rna}}^{\text{ctrl}}) \tag{5}$$

where each gene expression value is projected to a $d_{\text{gene}}$-dimensional embedding space. We apply $L$ layers of multi-head self-attention (MHA):

$$\mathbf{A}^{(l)} = \text{MHA}(\mathbf{E}^{(l-1)}, \mathbf{E}^{(l-1)}, \mathbf{E}^{(l-1)}) \tag{6}$$

$$\mathbf{E}^{(l)} = \text{LayerNorm}(\mathbf{E}^{(l-1)} + \text{FFN}(\text{LayerNorm}(\mathbf{E}^{(l-1)} + \mathbf{A}^{(l)}))) \tag{7}$$

The final RNA-seq features are obtained through attention-based pooling:

$$\mathbf{h}_{\text{rna}} = \sum_{i=1}^{G} \alpha_i \mathbf{E}_i^{(L)}, \quad \alpha_i = \frac{\exp(\mathbf{w}^T \tanh(\mathbf{W}_{\text{pool}} \mathbf{E}_i^{(L)}))}{\sum_{j=1}^{G} \exp(\mathbf{w}^T \tanh(\mathbf{W}_{\text{pool}} \mathbf{E}_j^{(L)}))} \tag{8}$$

Control cellular images are processed through a ResNet-style convolutional (He et al., 2016) architecture:

$$\mathbf{h}_{\text{img}} = \text{GlobalPool}(\text{ResNet}(\mathbf{x}_{\text{img}}^{\text{ctrl}})) \tag{9}$$

Drug conditioning information combines categorical and continuous variables:

$$\mathbf{h}_{\text{drug}} = \text{Fusion}([\mathbf{e}_{\text{compound}}, \mathbf{e}_{\text{cell}}, \text{Conc}(c_{\text{conc}}), \text{Time}(c_{\text{time}})]) \tag{10}$$

where $\mathbf{e}_{\text{compound}}$ and $\mathbf{e}_{\text{cell}}$ are learned embeddings for compound and cell line identities.

Knowledge graph integration enhances both molecular and genomic representations through structured biological knowledge from PrimeKG (Chandak et al., 2023). For drug embeddings, compounds are mapped to knowledge graph entities capturing molecular interactions, pathways, and pharmacological relationships. The heterogeneous graph neural network processes drug-protein, drug-drug, and protein-protein interactions:

$$\mathbf{h}_{\text{drug}}^{\text{kg}} = \text{KGDrugEncoder}(\mathbf{G}_{\text{drug}}, \mathbf{E}_{\text{rel}}) \tag{11}$$

where $\mathbf{G}_{\text{drug}}$ represents drug nodes and $\mathbf{E}_{\text{rel}}$ captures multi-relational edges. Similarly, gene expressions are enhanced with protein interaction networks and pathway information:

$$\mathbf{E}_{\text{RNA}}^{\text{kg}} = \text{KGGeneEncoder}(\mathbf{G}_{\text{gene}}, \mathbf{E}_{\text{ppi}}) \tag{12}$$

The knowledge graph embeddings are integrated additively with learned representations:

$$\mathbf{h}_{\text{drug}} = \mathbf{h}_{\text{drug}} + \alpha_{\text{drug}} \mathbf{h}_{\text{drug}}^{\text{kg}} \text{ and } \mathbf{E}_{\text{RNA}} = \mathbf{E}_{\text{RNA}} + \alpha_{\text{RNA}} \mathbf{E}_{\text{RNA}}^{\text{kg}} \tag{13}$$

where $\alpha_{\text{drug}} = 0.3$ and $\alpha_{\text{RNA}} = 0.3$ are learned weighting factors.

To capture cross-modal RNA-Image dependencies, we use multi-token cross-attention. Each modality is projected to $K$ token representations:

$$\mathbf{T}_{\text{rna}} = \text{RNAProj}(\mathbf{h}_{\text{rna}}) \qquad \mathbf{T}_{\text{img}} = \text{ImageProj}(\mathbf{h}_{\text{img}}) \tag{14}$$

Each modality goes through a self-attention block, then cross-attention is applied bidirectionally (Eq. 6):

$$\mathbf{T}_{\text{rna}}^{\text{cross}} = \text{MHA}(\mathbf{T}_{\text{rna}}, \mathbf{T}_{\text{img}}, \mathbf{T}_{\text{img}}) \qquad \mathbf{T}_{\text{img}}^{\text{cross}} = \text{MHA}(\mathbf{T}_{\text{img}}, \mathbf{T}_{\text{rna}}, \mathbf{T}_{\text{rna}}) \tag{15}$$

Enhanced features are obtained through residual connections and attention pooling:

$$\mathbf{h}_{\text{rna}}^{\text{enh}} = \text{AttentionPool}(\mathbf{T}_{\text{rna}} + \mathbf{T}_{\text{rna}}^{\text{cross}}) \qquad \mathbf{h}_{\text{img}}^{\text{enh}} = \text{AttentionPool}(\mathbf{T}_{\text{img}} + \mathbf{T}_{\text{img}}^{\text{cross}}) \tag{16}$$

The cross-modal features are combined with drug conditioning (Eq. 10) to form a unified representation:

$$\mathbf{h}_{\text{shared}} = \text{SharedEncoder}([\mathbf{h}_{\text{rna}}^{\text{enh}}, \mathbf{h}_{\text{img}}^{\text{enh}}, \mathbf{h}_{\text{drug}}]) \tag{17}$$

This shared representation captures the complex dependencies between molecular states, morphological features, and drug effects necessary for coherent multi-modal generation.

Treatment gene expression is generated through direct prediction from the shared representation (Eq. 17):

$$\mathbf{x}_{\text{rna}}^{\text{treat}} = \text{TranscriptomeHead}(\mathbf{h}_{\text{shared}}) \tag{18}$$

For image generation, we adapt rectified flow dynamics. Given noise $\mathbf{z}_0 \sim \mathcal{N}(0, \mathbf{I})$ and target image $\mathbf{x}_{\text{img}}^{\text{treat}}$, rectified flow defines a linear interpolation path:

$$\mathbf{x}_t = (1-t)\mathbf{z}_0 + t\mathbf{x}_{\text{img}}^{\text{treat}}, \quad t \in [0, 1] \tag{19}$$

The velocity field is defined as:

$$\mathbf{v}_t = \mathbf{x}_{\text{img}}^{\text{treat}} - \mathbf{z}_0 \tag{20}$$

and our multi-modal-conditioned UNet learns to predict this velocity:

$$\mathbf{v}_\theta(\mathbf{x}_t, t, \mathbf{h}_{\text{shared}}) \approx \mathbf{v}_t \tag{21}$$

The UNet incorporates cross-attention layers that attend to image conditioning derived from the shared representation:

$$\mathbf{c}_{\text{img}} = \text{ImageUNet}(\mathbf{h}_{\text{shared}}) \tag{22}$$

Our training strategy combines task-specific losses with cross-modal consistency objectives. We use a combination of MSE and auxiliary Pearson Correlation loss for transcriptome prediction:

$$\mathcal{L}_{\text{rna}} = 0.9 \cdot \text{MSE}(\mathbf{x}_{\text{rna}}^{\text{treat}}, \hat{\mathbf{x}}_{\text{rna}}^{\text{treat}}) + 0.1 \cdot \text{PC}(\mathbf{x}_{\text{rna}}^{\text{treat}}, \hat{\mathbf{x}}_{\text{rna}}^{\text{treat}}) \tag{23}$$

For rectified flow training, we minimize the velocity prediction error for image generation:

$$\mathcal{L}_{\text{img}} = \mathbf{E}_{t, \mathbf{z}_0, \mathbf{x}_{\text{img}}^{\text{treat}}} \left[ \|\mathbf{v}_\theta(\mathbf{x}_t, t, \mathbf{h}_{\text{shared}}) - (\mathbf{x}_{\text{img}}^{\text{treat}} - \mathbf{z}_0)\|^2 \right] \tag{24}$$

We implement triplet contrastive consistency (Stoica et al., 2025) to ensure well-aligned features produce better predictions than misaligned ones:

$$\mathcal{L}_{\text{triplet}} = \mathbf{E}[\max(0, \text{margin} - (\mathcal{L}_{\text{neg}} - \mathcal{L}_{\text{pos}}))] \tag{25}$$

where $\mathcal{L}_{\text{pos}}$ is the prediction error with aligned features and $\mathcal{L}_{\text{neg}}$ with misaligned features. The complete training objective combines all losses (Eqs. 23-25):

$$\mathcal{L}_{\text{total}} = w_{\text{rna}}\mathcal{L}_{\text{rna}} + w_{\text{img}}\mathcal{L}_{\text{img}} + w_{\text{triplet}}\mathcal{L}_{\text{triplet}} \tag{26}$$

where the weights are set to $w_{\text{rna}} = 0.5$, $w_{\text{img}} = 0.5$, $w_{\text{triplet}} = 0.05$.

Treatment RNA-seq is generated through a single forward pass using Eq. 18:

$$\mathbf{x}_{\text{rna}}^{\text{treat}} = f_\theta(\mathbf{x}_{\text{rna}}^{\text{ctrl}}, \mathbf{x}_{\text{img}}^{\text{ctrl}}, \mathbf{c}) \tag{27}$$

For high-quality image generation, we use an adaptive DOPRI5 solver (Dormand & Prince, 1986) that iteratively integrates the learned velocity field:

$$\frac{d\mathbf{x}}{dt} = \mathbf{v}_\theta(\mathbf{x}_t, t, \mathbf{h}_{\text{shared}}) \tag{28}$$

Starting from noise $\mathbf{x}_0 \sim \mathcal{N}(0, \mathbf{I})$ at $t = 0$, the solver adaptively adjusts step sizes based on error estimation to reach the treatment image at $t = 1$. The adaptive integration ensures both computational efficiency and generation quality. The DOPRI5 method uses a 5th-order Runge-Kutta scheme with embedded 4th-order error estimation for automatic step size control. The step size $h$ is adapted based on the estimated local truncation error to maintain tolerance levels.

A.4 IDENTIFYING GENES CHANGING MORPHOLOGIC PHENOTYPES:

We evaluated gene contributions to morphological recovery using gradient-based feature importance with respect to the flow-matching loss during inference. Through cross-modal embedding co-registration, the model correctly identified gene modules linked to treatment-induced morphology changes for example, apoptosis pathway genes activated in A549 cells under camptothecin or etoposide, and reduced activation of cell-cycle modules under proliferation inhibitors compared to negative controls. To map pathways affected by drug treatments, we extracted gene importance scores from the model's latent representations for each sample (Figure 11). Scores capture the model's learned associations between gene expression and drug-induced transcriptional changes. For each drug–cell line pair, we averaged scores across samples, selected the top 200 genes, and performed gene set enrichment analysis with GSEApy using the MSigDB Hallmark 2020 collection (adjusted p ¡ 0.25). This pipeline systematically linked drug-specific transcriptional signatures to biological processes, revealing both universal stress responses (e.g., EMT, TNF-alpha/NF-kB signaling) and cell line–specific activations. Enrichment results were visualized with scanpy-style dotplots, where dot size reflects gene overlap and color intensity indicates significance, enabling clear comparison of pathway activation across compounds and cell types.

Gene set enrichment analysis revealed distinct cell line-specific responses to pharmaceutical compounds, with notable differences in pathway activation between A549 lung cancer cells, human aortic smooth muscle cells (HASMC), and human dermal fibroblasts. A549 cells consistently showed limited pathway enrichment, with most drugs activating only TNF-alpha signaling via NF-kB and occasionally inflammatory response pathways. In contrast, both HASMC and dermal fibroblasts demonstrated robust, multi-pathway responses to the same compounds, suggesting that A549 cells may have inherent resistance mechanisms or altered sensitivity to drug-induced transcriptional changes. Epithelial Mesenchymal Transition (EMT) emerged as the most consistently enriched pathway across drug-cell line combinations, appearing as the top-ranked pathway in nearly all HASMC and fibroblast treatments. This universal EMT activation suggests that pharmaceutical stress triggers fundamental cellular reprogramming programs associated with cell plasticity and survival. TNF-$\alpha$ signaling via NF-$\kappa$B represented the second most common response, activated across all three cell lines, indicating that drug treatment consistently triggers inflammatory stress response cascades regardless of the specific compound mechanism of action.

Beyond the universal stress signatures, each cell type exhibited specialized pathway responses reflecting their distinct biological functions. HASMC consistently activated vascular-specific pathways including angiogenesis, coagulation, and hypoxia response, which aligns with their role in vascular homeostasis and their sensitivity to oxygen and hemodynamic stress. Human dermal fibroblasts uniquely enriched for tissue remodeling pathways including myogenesis, adipogenesis, and UV response, consistent with their role in tissue repair and their exposure to environmental stressors. Notably, fibroblasts also showed strong enrichment for interferon gamma response pathways, suggesting heightened immune surveillance capabilities compared to the other cell types. Our multimodal model's gene importance scoring successfully captured biologically relevant drug-target relationships, as evidenced by cell-type-specific pathway enrichment patterns. The A549 lung cancer cell line's dominant TNF-$\alpha$/NF-$\kappa$ B activation across diverse compounds aligns with established literature showing that KRAS-mutant lung adenocarcinomas exhibit heightened inflammatory signaling dependency. This universal inflammatory response contrasts with the diverse EMT and angiogenesis signatures observed in primary cells, recapitulating known differences between transformed cancer cells and stromal cells. Mechanistic validation was further demonstrated through drug-specific responses: DNA-damaging agents activated p53 pathways in TP53-intact A549 cells, while dabrafenib triggered compensatory KRAS signaling in non-mutant cells - mirroring clinical resistance mechanisms.

The model's biological accuracy extends to morphological predictions, as evidenced by paired Cell Painting data showing dramatic cell number decreases in samples treated with apoptosis-inducing drugs like nocodazole, consistent with the observed enrichment of apoptosis pathways in our transcriptional analysis. These concordant transcriptional and morphological responses demonstrate that our multimodal architecture captures functionally relevant biological relationships rather than mere correlative patterns, validating the gene importance scores as mechanistically informative features for drug response prediction. The enrichment patterns largely validated known drug mechanisms of action. DNA-damaging agents such as etoposide, mitoxantrone, and camptothecin consistently

activated p53 pathway and apoptosis responses in responsive cell lines. Anti-inflammatory compounds including dexamethasone and corticosterone showed expected modulation of inflammatory response and IL-2/STAT5 signaling pathways. Targeted inhibitors like dabrafenib demonstrated compensatory KRAS signaling activation, consistent with known resistance mechanisms in cancer cells. However, the limited response in A549 cells to many compounds suggests potential resistance mechanisms that may be clinically relevant for lung cancer treatment strategies.

These findings demonstrate that drug response profiling through transcriptional analysis can reveal both universal cellular stress responses and cell type-specific vulnerabilities, providing insights into both drug mechanism of action and potential therapeutic resistance patterns across different tissue contexts.

## A.5    FURTHER EXAMPLES OF GENERATED IMAGES

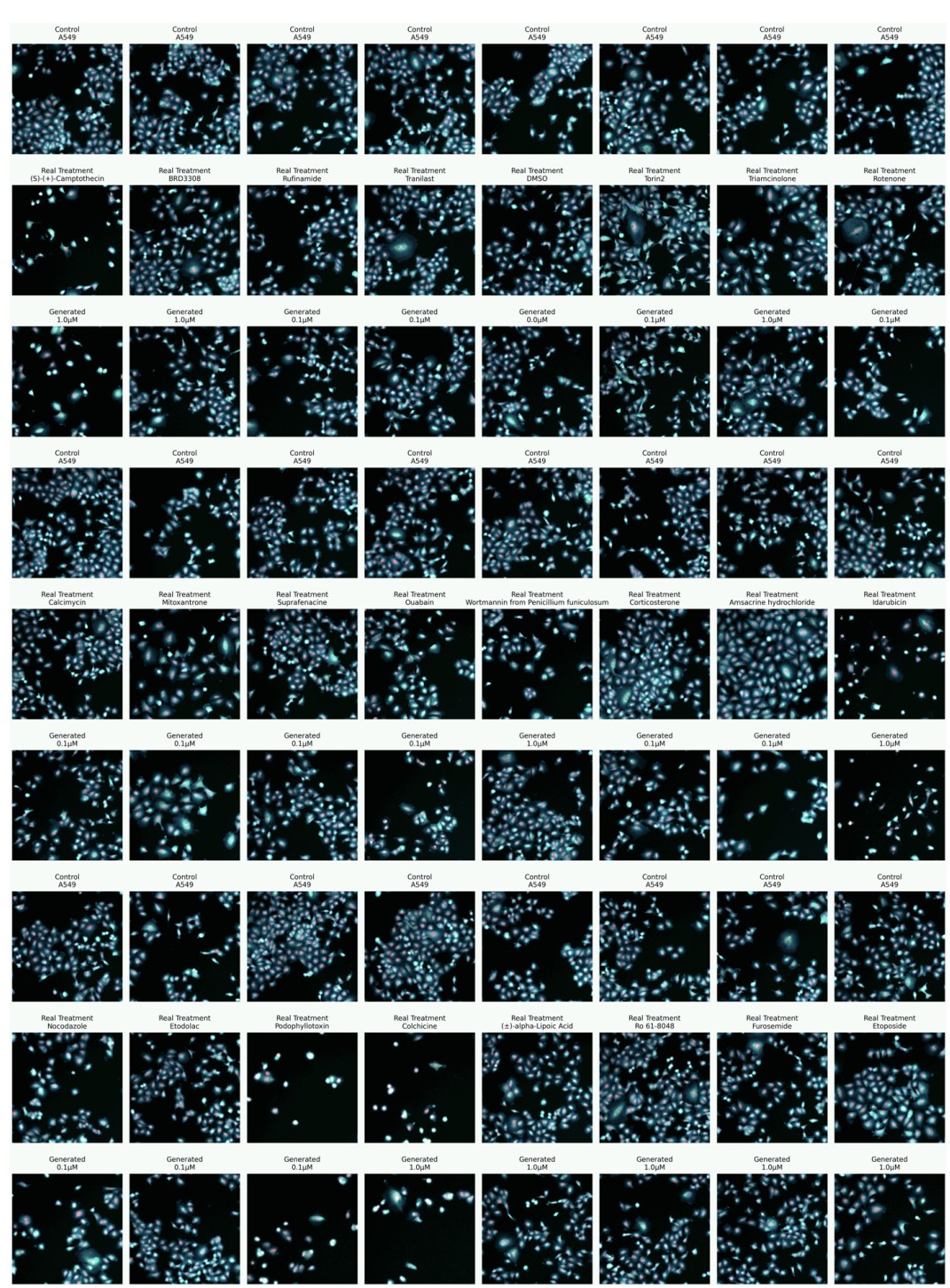

Figure 12: Cell line A549

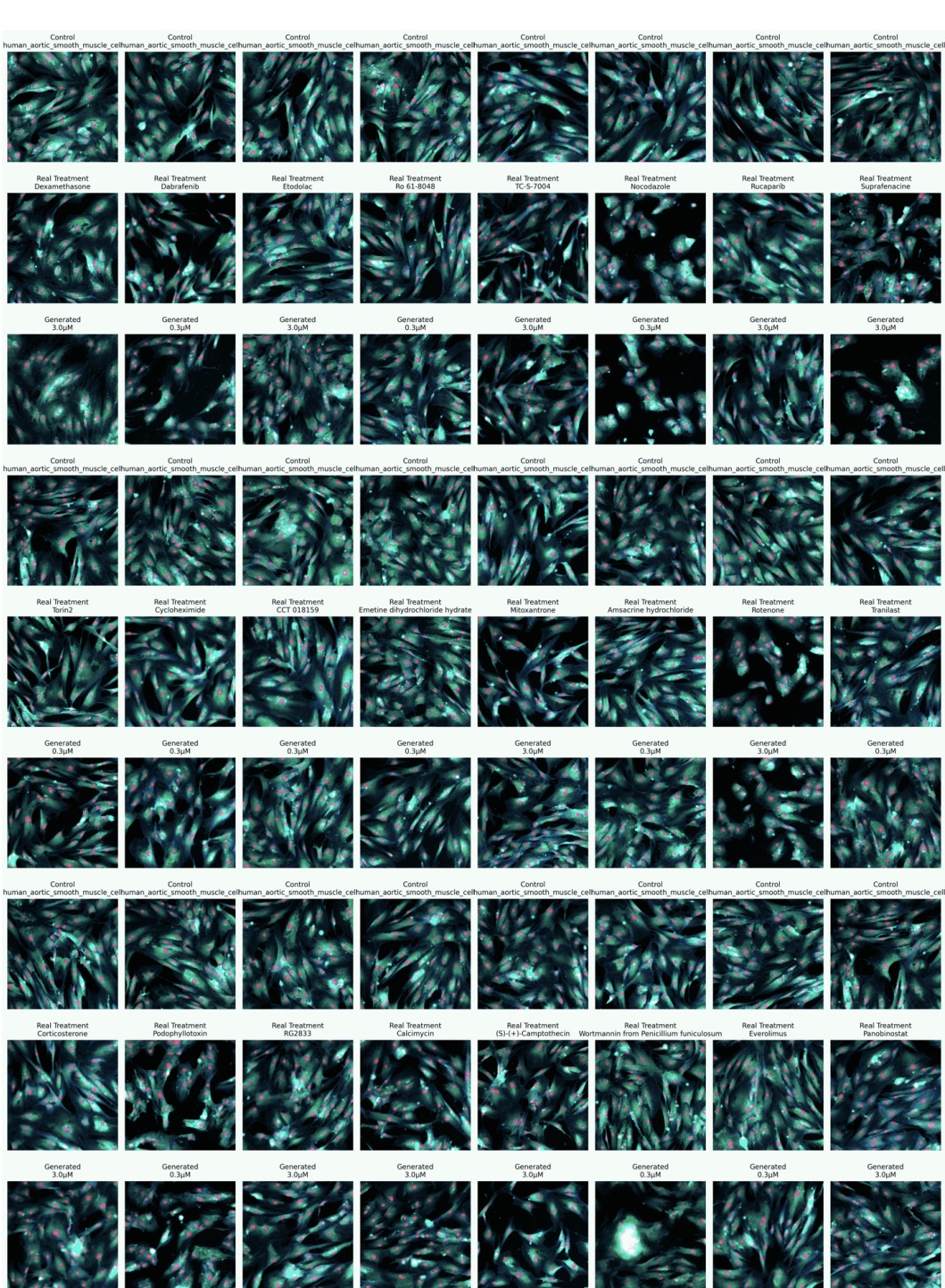

Figure 13: Cell line Aortic Smooth Muscle Cell

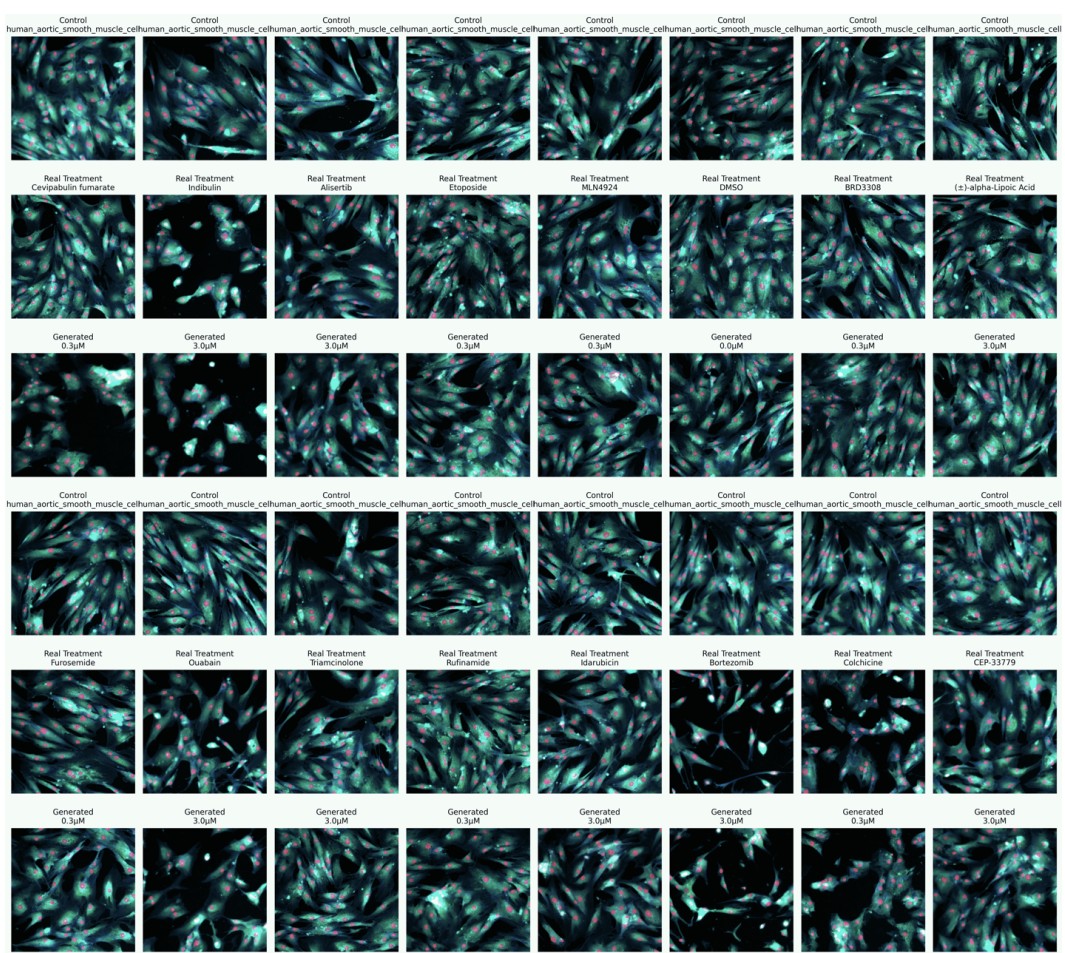

Figure 14: Cell line Aortic Smooth Muscle Cell

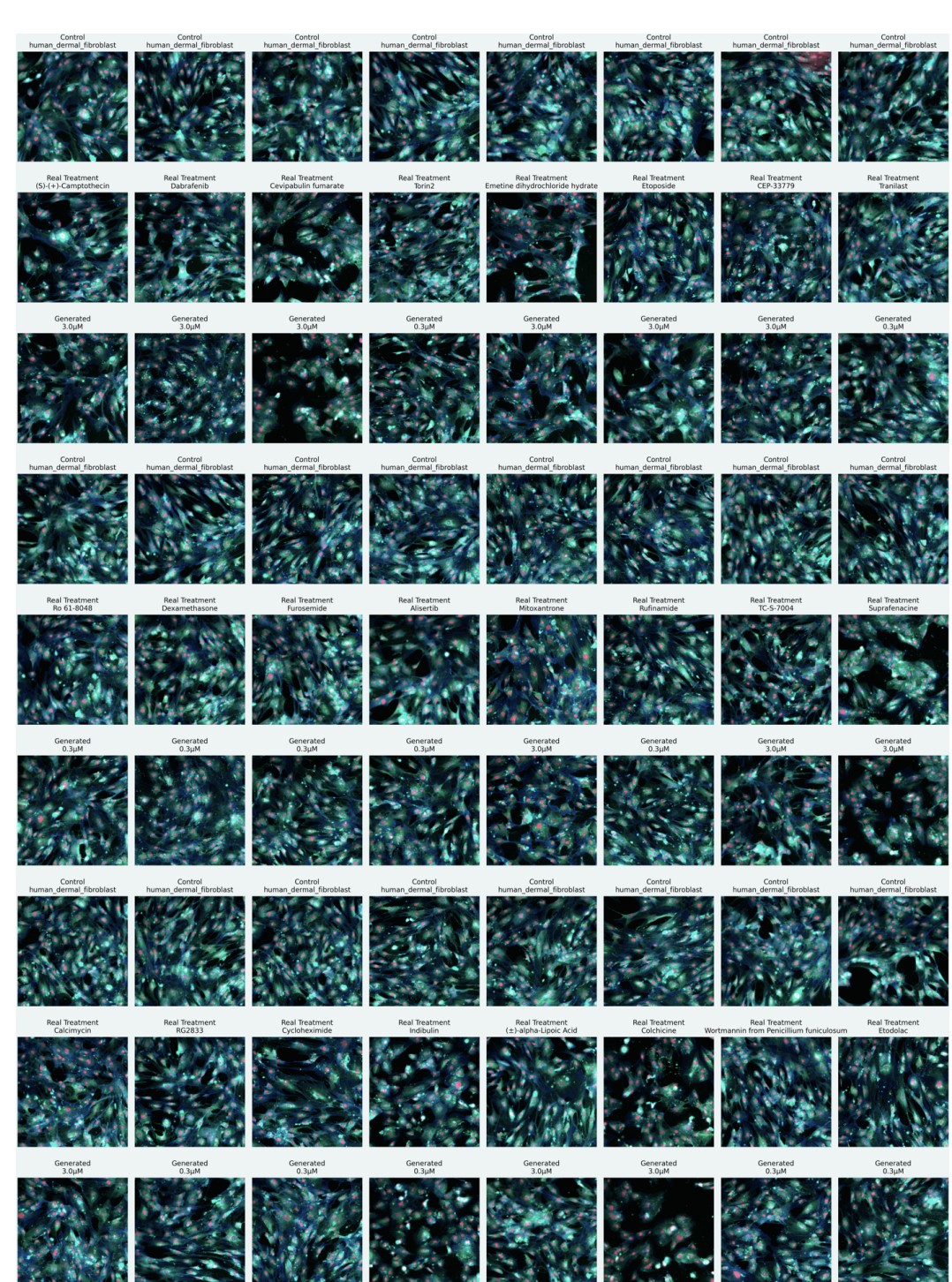

Figure 15: Cell line Dermal Fibroblast

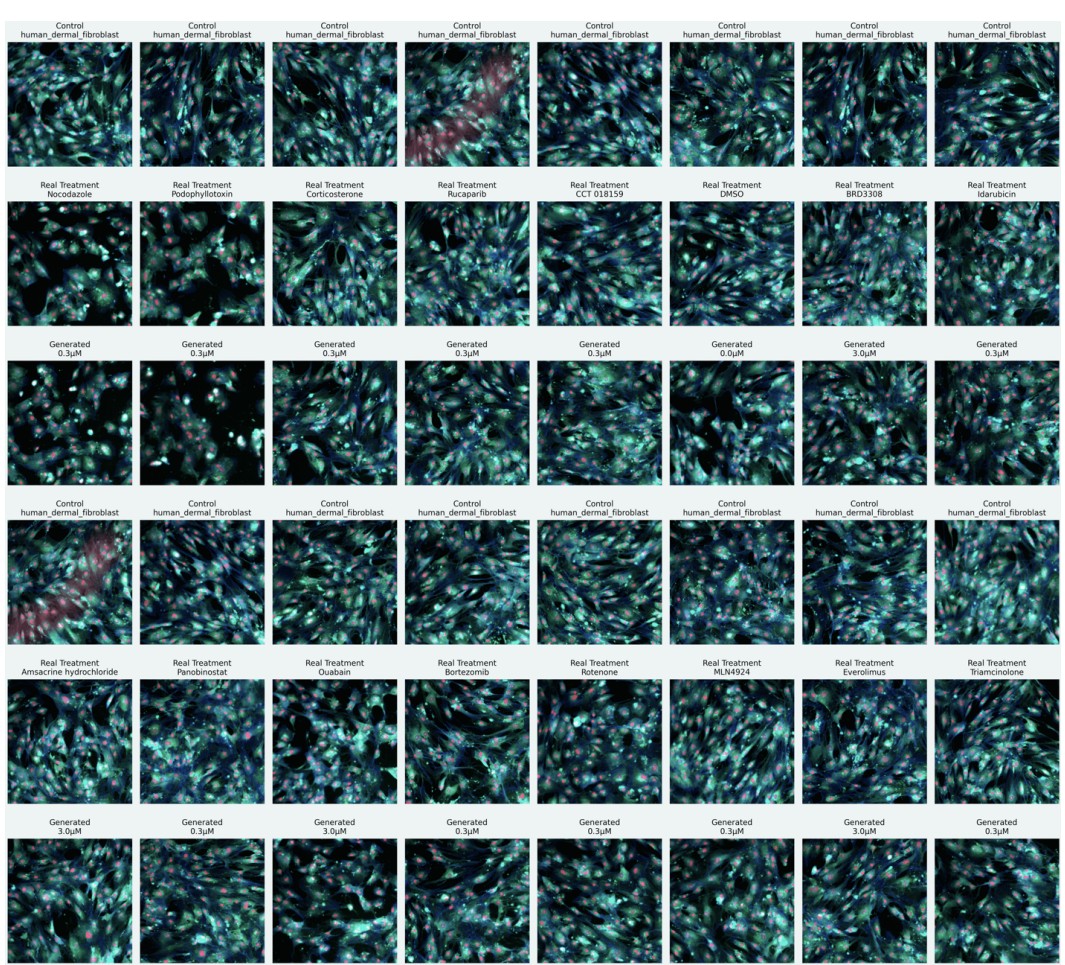

Figure 16: Cell line Dermal Fibroblast

