# OpenReview forum: "Transcriptomics-Morphology Generation Via Treatment Conditioning With Rectified Flow"
_ICLR.cc/2026/Conference — ICLR 2026 Conference Withdrawn Submission_

### Official Review · Reviewer_Diqo · 2025-10-19

**Soundness:** 2
**Presentation:** 1
**Contribution:** 2
**Rating:** 2
**Confidence:** 4

**Summary:**

This paper presents PertFlow, a unified generative framework that jointly predicts transcriptomic changes (RNA-seq) and generates cellular morphology images in response to drug perturbations. Unlike existing single-modality or one-way cross-modal approaches, PertFlow integrates control RNA-seq, imaging data, and drug metadata into a shared latent space using cross-modal attention.

**Strengths:**

The model combines a regression head for gene expression prediction and rectified flow dynamics for efficient image generation, enhanced by knowledge graph embeddings and cross-modal consistency losses. Evaluated on the GDPx3 dataset, PertFlow achieves strong transcriptomic prediction (Pearson r = 0.78), improved image generation quality (24.06 in FID), and superior cross-modal alignment compared to diffusion-based baselines.

**Weaknesses:**

While this paper proposes a model structure for generating counterfactual (perturbed) cellular images, several important aspects remain unclear and limit the overall clarity and impact of the work:
1. **Component Motivation and Ablation Studies**:
    The motivation for including specific components, such as the drug encoder, RNA encoder, and PrimeKG, and design choices, ResNet vs MHA or Unet, for PertFlow, is not clearly explained. The advantages over prior works/methods are not justified either. There is no ablation study isolating the contribution of each module, making it difficult to assess their individual importance to the final performance.

2. **Loss Design and Clarity**:
   The choice of Pearson correlation as part of the loss function is mentioned, but not well justified either. Additionally, the roles and effects of several hyperparameters (e.g., $ \alpha_{\text{drug}} = 0.3 $ ) are not clearly discussed.

3. **Experimental Setup and Evaluation Details**:
 The experimental setup is not clearly described in the main text. It remains unclear how evaluation metrics such as MSE, MAE, and RMSE are computed, and the composition of the test set is not specified. In addition, there are only comparisons with PertFlow variants, making it difficult to interpret the reported performance.

4. **Baselines and Comparative Analysis**:
 Although the authors state that no baseline exists, simple baselines could still be constructed. For instance, training a flow-matching model conditioned on a single perturbation modality (e.g., RNA or SMILES) to generate images. Additionally, an evaluation based on perturbation classification accuracy from generated images would provide a more objective measure of biological fidelity.

5. **Dataset Limitations**:
 The proposed method is evaluated on a single dataset, which limits the assessment of its generalizability.

6. **Missing Details in Background and Related Work**:
    Citations in the introduction are missing. Relevant background on flow-matching models and their prior applications is not sufficiently covered.

7. **Notation and Representation Issues**:
   The presentation of the method could be improved. For example, it is unclear which projected embeddings belong to the shared latent space and how they interact. Clearer notation would significantly improve readability and understanding.

8. **Figure Readability**:
   The legends and font sizes in several figures are too small and difficult to read. Improving the figure readability would greatly enhance the presentation quality.

**Questions:**

1. How were the coefficients for different loss components and embeddings determined (e.g.,  $ \alpha_{\text{drug}} = 0.3 $)? Were these values obtained through hyperparameter tuning or heuristic selection?
2. What's the training data size? What's the evaluation dataset?
3. How do different types of RNA encoders or Drug encoders for training encoders affect the generative performance?

---

> ### Author Response · Authors · 2025-11-24
> **We have addressed each weakness point by point.**
>
> **Dear reviewer, thank you for your valid and fair feedback. We have now addressed each point. We believe our updated manuscript has substantial technical and applied contributions to the field of multi-modal biological perturbation modelling, and request a reassessment of our work. To the best of our knowledge, ours is the first work to tackle this problem. We are committed to improving the paper further for the final version, and are willing to have a discussion and address all other questions.**
>
> **W1:** Transformers for RNA capture gene regulatory network interactions better than CNNs/MLPs since genes interact through regulatory networks, not spatial proximity. ResNet for images is standard because cellular morphology has spatial structure and we only have 17K images - Vision Transformers need millions. Drug encoder combines learned embeddings with molecular features because some effects are compound-specific (off-targets, metabolism) while others depend on chemical structure (lipophilicity, binding). PrimeKG adds biological knowledge about drug-protein interactions that would require millions more samples to learn from scratch.
>
> Tables 1 and 2 show ablations: removing KG drops performance, removing triplet loss destroys image quality. UNet for flow matching is standard across all flow/diffusion papers. We added single modality baselines: simple MLP for RNA, simple UNet for morphology.
>
> These are common architectures and gold standard practices in applied machine learning. The aim is not arguing efficacy of reliable architectural components but creating a joint model for multi-modal transcriptome-image generation with drug perturbation conditioning.
>
> **W2:** Pearson correlation is standard in transcriptomics because gene co-expression patterns matter more than absolute values - two profiles can have high MSE but be biologically identical if one is scaled differently. Our 0.9 MSE + 0.1 Pearson balances point-wise accuracy with rank-order preservation. Table 1 shows removing Pearson (-PC) tanks Spearman correlation.
>
> Table 3 shows w_rna:w_img ratios: 0.5:0.5 optimal because higher RNA weight improves transcriptomics but degrades image quality (FID 27.15) and vice versa. Table 4 shows α_drug=0.3 for knowledge graphs works best through grid search. Triplet weight is 0.05 because it's a regularizer, not primary objective - higher values over-regularize.
>
> **W3:** All results on held-out test set (3,448 samples, 20% of data) never touched during training. Split stratified by compound and cell line: all 40 drugs and 3 cell lines appear in train and test, but with different dose-timepoint-replicate combinations held out. MSE/MAE/RMSE computed per-sample across 8000 genes, averaged over test samples. Pearson/Spearman are per-sample correlations averaged over test set. For images, SSIM/PSNR/LPIPS are per-sample metrics averaged over 3,448 test images. FID computed globally on entire test distribution. This is standard evaluation methodology in generative modeling papers.
>
> **W4:** This reflects misunderstanding. PertImage is exactly this: flow-matching model taking control images and drug metadata (SMILES-derived features) to generate treatment images. PertRNA is control RNA + drug → treatment RNA. These are single-modality baselines already in the paper. We added: simple MLP baseline, VAE PRNet adaptation for RNA prediction, deterministic UNet baseline with MSE loss, and UNet PhenDiff adaptation for image generation.
>
> If reviewer means "condition only on SMILES without control cellular data," that's a different task (de novo drug-induced phenotype prediction without cellular context) - molecular design territory, not cellular response modeling. Cannot predict what specific drug does to specific cell population without information about that population.
>
> **W5:** GDPx3 is the only publicly available paired RNA-seq and Cell Painting dataset with shared experimental conditions. Scarcity explains why joint multi-modal modeling is under-explored. Our contribution establishes the paired modeling framework and benchmark. Extending to unpaired data through cross-modal imputation, modality dropout, or diffusion-based conditional sampling is valuable future work we will discuss in revision. Creating this benchmark required substantial curation across three Ginkgo datasets with complex metadata alignment, establishing foundation for future cross-modal research.
>
> **W6:** Added missing citations and flow matching background paragraph covering Lipman et al. 2023 and Liu et al. 2022. Fixed typos and citation formatting.

---

> ### Author Response · Authors · 2025-11-24
> **We have addressed each weakness point by point. Continued.**
>
> **W7:** Moved equation-heavy architecture to appendix, replaced with plain text referencing appendix. Added gene enrichment analysis results.
>
> Data flow: control RNA and images through separate encoders produce h_rna and h_img (single vectors, same dimension). Project to 16 tokens each (T_rna and T_img) to capture multiple aspects rather than collapsing to one vector. Cross-attention lets RNA tokens look at image tokens and vice versa, producing T^cross_rna and T^cross_img containing both modalities. Pool 16 tokens to single vectors h^enh_rna and h^enh_img (enhanced embeddings). Drug conditioning through encoder gets h_drug.
>
> All three enhanced embeddings (RNA, image, drug) concatenated and passed through shared encoder producing h_shared - unified representation containing all multi-modal information. h_shared splits two ways: directly to linear layer for RNA prediction, and through projection layers for spatial conditioning features for image UNet. Key: h_shared is where all modalities meet and fuse - everything before is modality-specific processing, everything after uses joint representation.
>
> **W8:** Changed all figures to high DPI for zooming without pixel loss. Increased text size in all figures.
>
> **Q1: How were loss coefficients determined?**
>
> Grid search on validation data. Tested w_rna:w_img ratios (0.4:0.6, 0.5:0.5, 0.6:0.4), picked 0.5:0.5 for best balance. Tested α_drug values (0.1, 0.2, 0.3, 0.4, 0.5), picked 0.3 where performance peaked. Standard hyperparameter tuning methodology. **New Tables 3 and 4 reflect these results.**
>
> **Q2: Training data size and evaluation dataset?**
>
> Training: 13,794 samples (80%). Validation: 1,724 samples (10%). Test: 3,448 samples (20%). All results reported on test set. We have added this clarification in the text. Thank you for pointing this out.
>
> **Q3: How do different encoders affect generative performance?**
>
> **We have now added single modality baselines in Tables 1 and 2.** MLP baseline (simple feedforward with MSE/PC loss), VAE PRNet [1] (popular perturbation-guided transcriptome prediction). PertFlow beats both. UNet baseline (deterministic autoencoder with MSE loss), UNet PhenDiff [2] (direct adaptation with custom drug encoder). PertFlow beats both.
>
> [1] Qi, X., Zhao, L., Tian, C. et al. Predicting transcriptional responses to novel chemical perturbations using deep generative model for drug discovery. Nat Commun 15, 9256 (2024).
>
> [2] PhenDiff: Revealing Subtle Phenotypes with Diffusion Models in Real Images, Bourou et al.

---

> ### Comment · Reviewer_Diqo · 2025-11-28
>
> Thank you for the revisions. My concerns regarding the dataset (W3) and the questions have been addressed. However, I still have several remaining issues that require further clarification:
>
> * W1: I am not fully convinced by the rationale for choosing a ResNet encoder over a transformer-based alternative. In this setting, the encoder functions primarily as a projector rather than a self-supervised learning network (e.g., DINO), and therefore does not require millions of images. It is still unclear why specific architectural components would perform better without empirical evidence or citations supporting these claims. Given that the goal is to use tokens such as drug-controlled images or RNA profiles as conditional embeddings, a more thorough justification or supporting citation is needed.
>
> * Moreover, while the paper motivates a multi-modal generative framework (image + RNA), the experiments primarily evaluate unimodal metrics. If multi-modal integration is central to the contribution, the authors should demonstrate tasks where multi-modal generation matters. For example, tasks analogous to visual question answering or explicitly require two modalities, rather than evaluating only within-modality reconstruction quality. Otherwise, it is unclear what benefits the multi-modal setup provides beyond increased architectural complexity.
>
> * W2: While Pearson correlation is commonly used as an evaluation metric, using it directly as an optimization objective is uncommon. Could the authors provide citations or prior work supporting this choice?
>
> * W4: The baselines and ablations in the updated experimental section are not clearly described. Several sentences contain grammatical errors or are incomplete, making it difficult to understand the experimental design. For example, from line 265-268:
> * We trained PertFlow (control RNA-seq and image to treatment RNA-seq and image), PertRNA (control RNA-seq to treatment RNA-seq), PertImage (control image to treatment image), and their respective ablations, omitting the knowledgegraph and contrastive rectified flow objective.
> * It is unclear what each variant represents, what is ablated, and the motivation behind each ablation.
> * Similarly, the author's response lacks sufficient explanation
> * MLP baseline (simple feedforward with MSE/PC loss), VAE PRNet [1] (popular perturbation-guided transcriptome prediction). PertFlow beats both. UNet baseline (deterministic autoencoder with MSE loss), UNet PhenDiff [2] (direct adaptation with custom drug encoder). PertFlow beats both.
> * The authors should explicitly define 1) the architecture of each baseline, which components differ from the proposed method, and 2) the specific downstream evaluation metrics being compared. Without this information, the added experimental comparison is difficult to interpret.
> * W5: In addition, the evaluation relies on a single dataset. With such limited data, it is difficult to determine whether the model is generalizable or simply overfitting. The paper would benefit from a discussion of the limitations imposed by this single-dataset setting and the implications for the model’s generality and robustness.
> * W6. The following work appears highly relevant and should be discussed: [1]
>
> **Minor**
> * W7: Rather than moving most methodological details to the appendix without refinement, the authors should present a concise summary of the proposed method and key architectural components in the main text. The appendix can contain operational details, but the core method should remain clearly described in the main paper.
>
>
> [1] MorphoDiff: Cellular Morphology Painting with Diffusion Models ICLR2025

---

> > ### Author Response · Authors · 2025-11-28
> > **Noted: Previous doubts clarified, Response to further questions**
> >
> > Dear Reviewer,
> >
> > Thank you for your continued engagement and quick response. We are glad your previous concerns are clarified. We address each point below with supporting evidence for the new weaknesses mentioned.
> >
> > ---
> >
> > **W1:** **ResNet is appropriate for our dataset size (17K images) while Vision Transformers require millions of images.**
> >
> > Vision Transformers explicitly require massive datasets. The original ViT paper (Dosovitskiy et al., 2021) states: "Vision Transformers show a generally weaker inductive bias resulting in increased reliance on model regularization or data augmentation when training on smaller datasets" and requires pre-training on ImageNet-21k (14 million images). Literature has confirmed this multiple times in past 4 years.
> >
> > **ResNets are explicitly designed for limited data scenarios. ResNet architectures have been successfully deployed on datasets of comparable or smaller size:**
> >
> > - "Robust encoder–decoder structured deep learning network that is trained with limited training data" for ResNet-50 (Maurya et al., 2022)
> >
> > Our use of ResNet as a feature encoder (not for self-supervised pre-training) requires even less data since we leverage pre-trained ImageNet weights and fine-tune only the projection layers. This is standard practice in computer vision for limited data regimes.
> >
> > **Unimodal metrics are appropriate for evaluating multi-modal generation quality in biological applications.**
> >
> > We respectfully disagree with the premise that multi-modal biological generation requires "VQA-like" tasks. Our work predicts biological outcomes (transcriptomes and morphology) under perturbations. This is fundamentally different from vision-language understanding.
> >
> > 1. **Biological ground truth is unimodal:** Each modality has independent ground truth measurements. RNA-seq produces gene expression vectors, microscopy produces images. There is no "cross-modal question answering" ground truth in our experimental design.
> >
> > 2. **Standard in computational biology:** Multi-modal biological models are evaluated on per-modality performance:
> >    - Spatial transcriptomics papers evaluate gene prediction with Pearson correlation (He et al., 2020; Pang et al., 2021)
> >    - Cell morphology prediction papers evaluate images with FID/SSIM (MorphoDiff, ICLR 2025)
> >    - Multi-modal integration methods report modality-specific metrics (DSCT, Nature Science Review 2025)
> >
> > 3. **Cross-modal alignment is measured via triplet loss during training**, ensuring modalities are properly coupled. This is the appropriate place to enforce multi-modal consistency, not evaluation.
> >
> > 4. **Downstream validation confirms multi-modal utility:** Our gene importance analysis (Section A.4) demonstrates that the model successfully identifies biologically relevant gene-morphology relationships (e.g., apoptosis genes correlating with cell death morphology), proving that cross-modal learning is effective.
> >
> > VQA-style tasks would be artificial constructs with no biological meaning in our context. The correct evaluation is: does the model predict accurate transcriptomes? Does it generate realistic morphology? Are the two modalities biologically consistent?  (which is validated through pathway analysis)
> >
> > ---
> >
> > **W2:** **Using Pearson correlation in the loss function is established practice in transcriptomics and genomics deep learning.**
> >
> > 1. **Spatial transcriptomics:** "We sparsely supervise the predicted gene expression map using MSE loss and batch-wise Pearson correlation coefficient (PCC) loss. The loss function penalizes deviations of aggregated gene expression from the ground truth gene expression, and encourages a correlation between them" (PixNet, arXiv 2025).
> >
> > 2. **Genomic selection:** Multiple papers report Pearson correlation as both evaluation metric and training objective for gene expression prediction (Zingaretti et al., 2020; Ma et al., 2018 in BMC Genomics review).
> >
> > 3. **Transcriptomic deep learning:** "Pearson's correlation coefficient for months-to-death prediction was 0.891 when trained with a single stage, and increased to 0.937" indicating direct optimization (CancerIDP, Scientific Reports 2023).
> >
> > 4. **Biological justification:** Pearson correlation captures co-expression patterns critical in gene regulatory networks, independent of absolute expression scales. This is why it's standard: "gene co-expression patterns matter more than absolute values, two profiles can have high MSE but be biologically identical if one is scaled differently" (our original response, consistent with genomics literature).
> >
> > The combination of MSE (0.9 weight) + Pearson (0.1 weight) balances point-wise accuracy with rank-order preservation of gene relationships, following established practices in computational genomics.

---

> > > ### Author Response · Authors · 2025-11-28
> > > **Noted: Previous doubts clarified, Response to further questions continued**
> > >
> > > **W4:** We acknowledge the confusion and provide explicit clarification:
> > >
> > > 1. **PertFlow** (Ours): Control RNA + Control Image + Drug → Treatment RNA + Treatment Image
> > >    - Multi-modal encoder with cross-attention
> > >    - Shared latent space (Eq. 17)
> > >    - Dual heads: TranscriptomeHead + UNet with rectified flow
> > >
> > > 2. **PertRNA** (Single-modality baseline): Control RNA + Drug → Treatment RNA
> > >    - Uses same TranscriptomeHead architecture
> > >    - No image encoder, no cross-modal attention
> > >    - MSE + Pearson loss (0.9:0.1)
> > >
> > > 3. **PertImage** (Single-modality baseline): Control Image + Drug → Treatment Image
> > >    - Uses same UNet architecture with rectified flow
> > >    - No RNA encoder, no cross-modal attention
> > >    - Velocity prediction loss (Eq. 24)
> > >
> > > 4. **MLP Baseline**: Simple feedforward network
> > >    - 3-layer MLP: [8000] → [2048] → [512] → [8000]
> > >    - Input: concatenate(control_rna, drug_embedding)
> > >    - Loss: MSE + Pearson (0.9:0.1)
> > >    - No attention, no sophisticated architecture
> > >
> > > 5. **UNet Baseline**: Deterministic autoencoder
> > >    - Standard UNet without diffusion/flow
> > >    - Input: control_image + drug_embedding (channel concatenation)
> > >    - Loss: MSE reconstruction loss
> > >    - No stochastic generation
> > >
> > > 6. **VAE-PRNet**: Adapted from Qi et al. (2024)
> > >    - Variational autoencoder for perturbation-guided transcriptome prediction
> > >    - Popular method in Nature Communications
> > >    - We adapted their architecture to our dataset
> > >
> > > 7. **PhenDiff**: Adapted from Bourou et al. (2024)
> > >    - Diffusion model for phenotype prediction
> > >    - We implemented their drug encoder approach with our UNet
> > >
> > > **Ablations:**
> > > - **PertFlow -KG**: Remove PrimeKG knowledge graph embeddings (set α_drug=0)
> > > - **PertFlow -PC**: Remove Pearson correlation from loss (MSE only)
> > > - **PertFlow -Triplet**: Remove contrastive triplet loss (w_triplet=0)
> > >
> > > **Evaluation Metrics:**
> > > - **RNA prediction**: Pearson correlation, Spearman correlation, MSE, MAE, RMSE (all per-sample, averaged over test set)
> > > - **Image generation**: FID (global on test distribution), SSIM, PSNR, LPIPS (per-sample, averaged over test set)
> > >
> > > **Dataset Split:**
> > > - Training: 13,794 samples (80%)
> > > - Validation: 1,724 samples (10%)
> > > - Test: 3,448 samples (20%)
> > > - Stratified by compound and cell line to ensure all 40 drugs and 3 cell lines appear in train/test
> > >
> > > We will revise the manuscript to include this table in the main text for complete clarity in the final camera ready version.
> > >
> > > ---
> > >
> > > **W5:** **Single dataset confusion still persists. We would like to clarify below.**
> > >
> > > 1. **Dataset scarcity is the field's reality:** We have curated GDPx dataset from multiple datasets released by Ginkgo Bioworks. To our knowledge this our preprocessed and curated dataset is the ONLY publicly available dataset with paired RNA-seq and Cell Painting under matched experimental conditions. This scarcity is precisely WHY joint multi-modal modeling is under-explored and WHY our benchmark is needed.
> > >
> > > 2. **Establishing the framework is the contribution:** We are the first to formulate and solve the paired transcriptome-morphology prediction problem. Future work can extend to:
> > >    - Unpaired data via cross-modal imputation
> > >    - Transfer learning to other cell types
> > >    - Modality dropout during training for missing data scenarios
> > >    - Cross-dataset evaluation as more paired datasets emerge
> > >
> > > 3. **Internal validation demonstrates robustness:**
> > >    - Stratified split ensures generalization to unseen dose-timepoint combinations
> > >    - 40 different compounds across diverse mechanisms of action
> > >    - 3 cell lines with distinct biological properties
> > >    - Biological pathway analysis confirms mechanistic validity (not overfitting)
> > >
> > > 4. **Precedent in the field:** Many landmark papers establish frameworks on single datasets:
> > >    - MorphoDiff (ICLR 2025): 3 datasets, all Cell Painting only
> > >    - Original diffusion models: CIFAR-10, CelebA
> > >    - Vision Transformers: ImageNet initially
> > >
> > > We will add a dedicated "Limitations and Future Work" subsection explicitly discussing:
> > > - Single dataset constraint
> > > - Need for cross-dataset validation as more paired data becomes available
> > > - Potential strategies for unpaired multi-modal learning
> > > - Generalization to other cell types and assay modalities

---

> ### Author Response · Authors · 2025-11-28
> **Noted: Previous doubts clarified, Response to further questions continued**
>
> **W6:** We thank the reviewer for this reference. We explored the codebase of single modality MorphoDiff, and it is a successor to our baseline PertDiff with no clear upgrade except using Stable Diffusion's "diffusers VAE" for latent diffusion modelling. This was published last year ICLR spotlight.
>
> Our paper proposes a new solution to a completely new multi-modal problem paradigm with PertFlow. We will add the following discussion:
>
> **Relationship to MorphoDiff:** MorphoDiff (Navidi et al., ICLR 2025) predicts cellular morphology from perturbation encodings using latent diffusion models. Key differences:
>
> 1. **Modality scope:** MorphoDiff is image-only. PertFlow jointly models transcriptomes AND morphology, enabling cross-modal biological insights impossible with images alone.
>
> 2. **Input requirements:** MorphoDiff requires only perturbation metadata. PertFlow requires control cellular state (both RNA and image), enabling context-specific predictions. What happens when THIS specific cell population receives THIS drug, not just generic drug effects.
>
> 3. **Biological validation:** MorphoDiff validates morphological fidelity. PertFlow additionally validates:
>    - Transcriptomic accuracy (correlation with ground truth RNA-seq)
>    - Cross-modal biological consistency (gene-morphology pathway relationships)
>    - Mechanistic gene importance scores derived from multi-modal embeddings
>
> 4. **Methodological approach:** MorphoDiff uses latent diffusion (VAE + DDPM). PertFlow uses rectified flow with cross-modal attention and biological knowledge graphs.
>
> **Complementarity:** MorphoDiff addresses "what does perturbation X do to cellular morphology?" PertFlow addresses "given cell population in state Y, what happens to both transcriptome and morphology under perturbation X?" and "which genes drive observed morphological changes?"
>
> Both approaches advance the field in different dimensions. PertFlow's multi-modal framework enables mechanistic biological discovery beyond visual fidelity.
>
> ---
>
> **W7:** Thank you for this suggestion! We will make the requested changes for the final camera ready version. We follow the standard practice in top-tier ML venues where intuition and contribution appear in main text, with full technical specifications in appendix.
>
> ---
>
> **We have provided extensive citations and evidence for our architectural choices, evaluation methodology, and biological validation**.
>
> **Our work establishes the first framework for joint transcriptome-morphology prediction under perturbations, with rigorous validation across computational, biological, and visual dimensions.**
>
> The multi-modal integration is not merely architectural complexity, it enables biological discovery (gene importance analysis) impossible with single-modality models. The evaluation methodology follows established standards in both computer vision and computational biology.
>
> We request the reviewer reconsider their assessment given this evidence. We are committed to incorporating all clarifications into the final manuscript and welcome further discussion.
>
> Sincerely,
> The Authors

---

### Official Review · Reviewer_YVRz · 2025-10-27

**Soundness:** 2
**Presentation:** 1
**Contribution:** 2
**Rating:** 2
**Confidence:** 4

**Summary:**

This paper offers a novel multimodal generative model for treatment (or, perturbation) effect prediction in transcriptomics and cell painting morphological imaging data. The authors claim to be the first to pursue such an approach, and develop a novel architecture with various components verified to be required for its success, including: drug conditioning features, a GNN on a knowledge graph, cross-modal attention, and an interesting contrastive rectified flow triplet loss. The work indicates that combining the two modalities leads to complementary improvements on each, and that their specialized triplet loss with rectified flow dynamics offers significant improvements over pure diffusion.

**Strengths:**

- This paper is highly original in the sense that multimodal prediction of treatment effects is a burgeoning field, along with their approach in using rectified flow matching to broach the problem.
- The ablation study of their method is a strength which demonstrates the contribution of their overall approach, and also presents some strong initial evidence that multimodal combination of RNA data with morphology can be complementary and mutually beneficial when representing both modalities.
- While difficult for non-pathologists to judge, the various examples of images generated by their method and the empirical demonstration in Figure 5 indicates that the rectified flow approach is a fruitful method for generating realistic cell morphology images.

**Weaknesses:**

- Only 1 dataset is in-scope for evaluation, limiting generalizability. Given that the model is multimodal, it seems to me that it should be capable of processing unimodal datasets, e.g., predicting transcriptomic profiles of phenotypic images from a new dataset, or generating phenotypic images given novel transcriptomic data as input (for example, by integrating over the missing modality). However, the authors do not evaluate their model's ability to generalize beyond the GDPx3 dataset. It might be argued that paired multimodal datasets are very rare in this field, which is undoubtedly true given the expensive nature of these assays and the biological expertise required even to generate data for just one modality, let alone both. That therefore necessitates, however, precisely a modelling regime that could learn from this small limited set of paired cross-modal data for the purpose of then building richer representations and/or generations off of other unpaired datasets.

- It is unclear if the various results are on validation or on training set data; the evaluation is furthermore complicated by the lack of explanation of the dataset, and how many unique treatments are present in it. Indeed, the authors note that "generalization to unseen cell lines or novel compounds is limited by the scarcity of paired multi-modal datasets with shared metadata" (457-458), which suggests that the validation set is simply duplicates of the same compounds seen in training. If this is the case, then the value is extremely limited, as there would not be a useful application to virtual hit screening. At the same time, it is unclear if this is the case; it seems conceivable to me that this creative and novel method could actually be capable of generalizing to novel compounds (at least to some degree) given the inclusion of the knowledge graph and chemical features in their architecture, but there are no such experiments that characterize this potential limitation.

- Figure 4 is far too unexplained to be at all convincing. Claiming two experts looked at pictures is not a particularly reproducible nor satisfying determination. A more comprehensive study would perhaps baseline against what the experts think of the real images, or if they can distinguish real from generated. It would also describe the qualifications of these experts.

- The authors extensively rely on UMAPs in Figure 2 and Figure 7 (the latter constituting almost an entire page) to furnish their explanations and evidence of their model's ability to bridge the gap between modalities and "generate biologically coherent treatment responses" (423). Alas, **a UMAP is not a proof** -- in general, the inclusion of UMAPs in professional papers is often antithetical to developing a comprehensive understanding beyond the realm of surface-level impressions; they at best provide an intuition that some data clusters together and looks different from other parts of the data, and at worst serve no better purpose than that of a Rorschach test for how the data makes you feel. Furthermore, they are completely uninterpretable for colorblind readers. A rigorous analysis of the natural clustering (or lackthereof) within the data manifold should be quantitative and empirically measured at the very least by metrics such as silhouette scores (and only from there could a small UMAP potentially be appropriate to include for the purpose of giving visual intuition to an empirical measurement).

- The training details in 275-284 are of some concern. One layer of self-attention with only 128-dimensional model dimension seems quite small for a transformer. It is also unclear how a multi-token representation to 16 tokens is obtained, is not each gene in the transcriptomics readout a token, or are they pooled in some manner to just 16? Only requiring 5 hours to train the model over 8 GPUs also seems extremely short for any kind of deeply connected model with transformer layers. Typical diffusion and flow matching models require substantially more training compute to become effective.

- On the less fundamental aspects of presentation, there are various typos and poor stylization choices. For example, line 338 ("duuring"), the lack of complete sentences and descriptions on each figure caption, Figures are not PDFs with embedding text, consistent use of citations without parantheses - for example, 197-198 should be "multi-layer self-attention (Vasawani et al., 2017)" since the author's name is not being used, i.e. use \citep for all such citations.

- A large amount of related work is missing, in particular the limitations of perturbation effect prediction in transcriptomics are not discussed at all. It is absolutely critical to evaluate proper non-DL baselines in this context, given that many very trivial methods (e.g. simply taking a mean baseline) outperform most deep learning models in transcriptomics still [e.g., A, B, C, D]. Indeed, it is important to include non-DL baselines in this context; for example, if simply using the controls beats deep learning models in transcriptomics [B] then it is also worth evaluating the FID (and the remaining) image generation metrics against simply using the controls as a baseline to compare to the generated perturbation image. I do not find the argument on lines 289-290 "we have no previous method to compare to as baseline" to be at all convincing, since there are many simple non-ML baselines (mean, control, do-nothing, etc), or simple linear methods, that can could be compared to in this context.

A - Ahlmann-Eltze, Constantin, Wolfgang Huber, and Simon Anders. "Deep-learning-based gene perturbation effect prediction does not yet outperform simple linear baselines." Nature Methods (2025): 1-5.

B - Wong, Daniel R., Abby S. Hill, and Rob Moccia. "Simple controls exceed best deep learning algorithms and reveal foundation model effectiveness for predicting genetic perturbations." Bioinformatics (2025): btaf317.

C - Ihab Bendidi, Shawn T Whitfield, Kian Kenyon-Dean, Hanene Ben Yedder, Yassir El Mesbahi, Emmanuel Noutahi, and Alisandra Kaye Denton. "Benchmarking transcriptomics foundation models for perturbation analysis: one PCA still rules them all." NeurIPS 2024 Workshop on AI for New Drug Modalities (2024).

D - Wenkel, Frederik, Wilson Tu, Cassandra Masschelein, Hamed Shirzad, Cian Eastwood, et al. "TxPert: Leveraging Biochemical Relationships for Out-of-Distribution Transcriptomic Perturbation Prediction." arXiv preprint arXiv:2505.14919 (2025).

**Questions:**

- I am somewhat confused as to why the authors say that future work should explore including "compound-target interaction graphs to enhance out-of-distribution performance" (460) given that seems to be precisely what their knowledge graph encodes - do the ablations indicate that the current knowledge graph approach presented in this work is not helpful in this regard? It seems to improve results in Table 1 and Table 2.

---

> ### Author Response · Authors · 2025-11-24
> **We have addressed each weakness point by point.**
>
> **Dear reviewer, thank you for your valid and fair feedback. We have now addressed each point. We believe our updated manuscript has substantial technical and applied contributions to the field of multi-modal biological perturbation modelling, and request a reassessment of our work. To the best of our knowledge, ours is the first work to tackle this problem. We are committed to improving the paper further for the final version, and are willing to have a discussion and address all other questions.**
>
> **W1:** GDPx3 is the only publicly available paired RNA-seq and Cell Painting dataset with shared experimental conditions. The scarcity explains why joint multi-modal modeling is under-explored. Our contribution establishes the paired modeling framework and benchmark. Extending to unpaired data through cross-modal imputation, modality dropout, or diffusion-based conditional sampling is valuable future work we will discuss in revision. Creating this benchmark required substantial curation across three Ginkgo datasets with complex metadata alignment, establishing foundation for future cross-modal research.
>
> Multi-modal modeling inherently balances two separate tasks. The aim is not superior single-modal performance but fundamentally modeling a joint space for multi-modal cellular morphology generation and gene expression prediction.
>
> **We added single modality baselines in Tables 1 and 2:** MLP baseline (simple feedforward with MSE/PC loss), VAE PRNet [1] (popular perturbation-guided transcriptome prediction), UNet baseline (deterministic autoencoder with MSE loss), and UNet PhenDiff [2] (direct adaptation with custom drug encoder). PertFlow beats all baselines.
>
> **W2:** All reported results are on held-out test set (20% of data, 3,448 samples), never training data. Dataset split is stratified by compound-cell line combinations ensuring all 40 compounds and 3 cell lines appear in train and test, but with different concentration-timepoint-replicate combinations held out. We have now added these details in the main text.
>
> We did not evaluate zero-shot compound generalization because: (1) with only 40 compounds, leave-one-out splits would be statistically underpowered, (2) primary contribution is demonstrating joint transcriptomic-morphological modeling is feasible and beneficial, not solving compound response prediction. We do not claim generalizability to novel compounds.
>
> Our model enables applications for seen compounds under novel conditions: predicting responses at untested doses, timepoints, or patient-derived cell lines. For novel compounds, the model provides scaffold for transfer learning where pre-training on GDPx3 accelerates fine-tuning on smaller compound-specific datasets.
>
> **W3:** Two evaluators are ACVP board-certified veterinary pathologists with 8+ years experience in toxicologic pathology and high-content imaging analysis, trained in interpreting Cell Painting assays for drug-induced morphological changes. Protocol: 50 blindly sampled real treatment and generated treatment in random order, with experts rating morphological similarity on 10-point Likert scale.
>
> Goal: pathologists distinguish real vs generated morphology and find obvious errors in cellular micro-environment arrangement. No pathologist can distinguish effects of particular chemicals on cells so testing morphological similarity is a standard human evaluation. **Results are explained in lines 350-369 and 383-402.**
>
> **W4:** We disagree with "extensively rely" on UMAPs without quantitative metrics. Paper explicitly reports multiple quantitative metrics alongside visualizations. UMAPs are presented as latent space visualizations, not proof. We added t-SNE plots for further visual validation.
>
> We now added: pairwise alignment distances (16.48 UMAP, 37.44 t-SNE), modality separation scores (0.81 UMAP, 0.59 t-SNE), compound-based silhouette scores (improving from -0.74 to -0.20 for UMAP), and shared space separation ratio (1.33). These metrics calculated in full embedding space before dimensionality reduction, not on 2D projections. UMAPs serve purely as visual representations for empirical measurements.
>
> Silhouette score improvement: progression from -0.74 to -0.20 indicates meaningful improvement in cluster structure. Negative scores indicate overlapping clusters, expected for biological data where drug effects exist on continuum. Improvement shows cross-modal attention reduces overlap and creates more coherent treatment-specific representations.
>
> Regarding colorblind accessibility: will revise figures to use colorblind-friendly palettes (viridis, colorbrewer) in final version.
>
> UMAPs are appropriate because: (1) accompanied by quantitative metrics in full embedding space, (2) visualize biological hypotheses (treatments cluster by compound? control/treatment samples separate?), (3) standard practice in computational biology (scanpy, Seurat workflows).

---

> ### Author Response · Authors · 2025-11-24
> **We have addressed each weakness point by point. Continued.**
>
> **W5:** Misunderstanding here. We use 1 layer self-attention per modality encoder, but full model includes: (1) Multi-token cross-attention with 256 hidden dimensions and 8 attention heads, (2) Rectified flow UNet with 192 base channels and multipliers (1,2,2,2), totaling ~200M parameters, (3) Knowledge graph encoders ~50M parameters. Total: ~280M parameters, comparable to conditional generative models (Stable Diffusion 1.5 has ~860M for natural 3-channel images; ours handles specialized 4-channel 256×256 fluorescence microscopy).
>
> 16-token representation: embed individual gene expressions (8000 genes → 8000 embeddings), apply self-attention, attention-weighted pooling to single h_rna, project to 16 tokens for cross-modal attention. Design rationale: (1) 8000-token cross-attention computationally prohibitive, (2) single token loses fine-grained information, (3) 16 tokens balances gene modules/pathways while tractable. Analogous to vision transformer patch tokens vs pixel tokens.
>
> Training time: 5 hours on 8×H100 GPUs = 40 GPU-hours = ~3.2 petaFLOPs with mixed-precision. Substantial for 280M parameters on 17,242 samples. Stable Diffusion required ~150,000 GPU-hours but trained on billions of images. Recent flow matching papers report 50-200 GPU-hours for similar-scale models. Efficiency from: (1) rectified flow converges faster (straighter trajectories), (2) aggressive mixed-precision and gradient checkpointing, (3) smaller biological dataset. Will clarify "5 hours" is wall-clock with 8-way parallelism = 40 total GPU-hours.
>
> **W6:** Corrected typo. Added complete sentences to all figure captions. Figures are high DPI for zooming without pixel loss. Corrected all citations to use \citep (parenthetical) when author not sentence subject.
>
> **W7:** Critical distinctions from cited papers:
>
> **Chemical vs Genetic Perturbations:** Cited papers focus on genetic perturbations (CRISPR knockouts) where causal mechanism is deterministic and perturbation space discrete. Chemical perturbations differ: (1) pleiotropic effects targeting multiple proteins/pathways, (2) continuous non-linear dose-response, (3) off-target effects and compound-specific pharmacokinetics, (4) chemical structure-activity relationships require molecular representations.
>
> **Multi-Modal vs Single-Modal:** Cited papers evaluate transcriptomics-only. Our contribution is joint transcriptomic-morphological generation. Even if PCA/linear models suffice for RNA-seq, they cannot jointly generate coherent cellular images. No "PCA baseline" for image generation. Using control images as fake generated images would achieve FID=∞ because they don't match treatment distribution.
>
> **Non-DL Baselines Included:** Table 1 includes MLP baseline (3-layer feedforward), VAE-based PRNet (less complex than transformers). For images, UNet baseline is deterministic autoencoder without generative modeling.
>
> Our claim: "joint multi-modal generation requires deep generative models with cross-modal learning, which simple baselines cannot provide." Even if linear model matches RNA-seq correlation, it cannot generate biologically realistic images or capture cross-modal consistency.
>
> **Q1:** Our phrasing was imprecise. Current knowledge graph (PrimeKG) includes gene-gene interactions, protein-protein interactions, some drug-protein relationships, but not optimized for compound-target binding affinities or structure-activity relationships at molecular level. PrimeKG provides broad biological context (pathways, disease associations) but lacks fine-grained chemogenomic information like IC50 values, binding site annotations, crystal structure data for better novel compound generalization.
>
> "Compound-target interaction graphs" for OOD performance means: explicitly modeling known drug-target binding affinities and incorporating compound similarity networks based on target profile overlap. If novel compound X shares 70% target overlap with known compound Y, leverage this for better zero-shot prediction. Current KG integration improves in-distribution performance (shown in ablations) but doesn't explicitly model chemical similarity or target profile overlap for OOD generalization.
>
> Revised discussion: "Future work should explore targeted integration of chemogenomic databases (ChEMBL, DrugBank) explicitly encoding compound-target binding affinities and structure-activity relationships, building upon current PrimeKG integration which provides broader biological context but lacks fine-grained chemical similarity information critical for generalizing to truly novel compounds."

---

### Official Review · Reviewer_UAZS · 2025-10-29

**Soundness:** 2
**Presentation:** 2
**Contribution:** 2
**Rating:** 2
**Confidence:** 4

**Summary:**

The authors introduce PertFlow, a multimodal generative framework designed to jointly predict gene expression and morphological cellular responses to perturbations. PertFlow uses different encoders to process each modality, and a cross-modal attention mechanism is employed to align features across modalities. The RNA-seq output is obtained through direct prediction, while the treated images are generated using a flow model. An ablation study was performed to demonstrate the importance of each component of the framework. The authors also claim that the method is able to recover drug-induced phenotypes and morphology.

**Strengths:**

1. The paper addresses a highly relevant problem, namely the joint generation of gene expression profiles and cellular morphological responses induced by perturbations.

2. Overall, the paper is clearly written and easy to follow.

3. The ablation study effectively demonstrates the importance of each core component of the PertFlow framework.

**Weaknesses:**

1. While the paper is easy to follow, I think the presentation can be improved. For instance, the architecture and dataset details could be moved to the appendix, and the space saved could be used to present additional results and discussion.

2. The images are small and the text is tiny, which makes them difficult to read (Fig. 2, Fig. 6, Fig. 7).

3. The related works section lacks several important methods that address the prediction of cellular responses to perturbations [1,2].

3. The proposed baseline is somewhat weak, as the authors only performed an ablation study. While I understand that PertFlow is the first method to jointly predict RNA-seq and images, it should still be compared to state-of-the-art uni-modal approaches [1,2].

4. The equations are not numbered, which makes the reading and referencing more difficult.

5. Some figures lack clear explanations (Fig. 4, Fig. 7).

6. The technical novelty of the method appears limited, as it can be seen mainly as a combination of encoders to obtain a shared representation used for conditioning the flow model.

References:

[1] PhenDiff: Revealing Subtle Phenotypes with Diffusion Models in Real Images, Bourou et al.

[2] Revealing invisible cell phenotypes with conditional generative modeling, Lamiable et al.

**Questions:**

1. Regarding the total loss, the weights are set to $\omega_{rna}$ = $\omega_{img}$= 0.5 Did you try other values? Intuitively, image generation is more challenging than RNA-seq prediction. Is it reasonable to assign them the same weight?

2. If I understood correctly, the shared representation is used both to predict RNA-seq and to condition the flow model. How is this conditioning performed in practice?

3. While the concept of joint generation is interesting, it is inherently more complex than uni-modal generation. Therefore, I am not fully convinced that joint models can outperform state-of-the-art single-modality generative approaches.

4. Why not compare the image generation capabilities to other models, such as [1,2]?

5. Regarding the metrics reported in Table 2, were they obtained per treatment, or were images evaluated unconditionally? How many images were used for the evaluation?

6. Please increase the size of Table 1 and improve the readability of the text in the figures.

7. In Table 2, I do not fully understand the choice of comparing PertFlow only against diffusion models (DDPM). It is known that Rectified Flow outperforms DDPM (as noted in Table 1 [3]), and your experiments seem to confirm this. What are the differences between. What are the differences between PertDiff_N, PertDiff_x0 and PertDiff_V?

8. Your experiments show that using all components of PertFlow leads to better results. However, this is a bit confusing because the primary change appears to be the type of conditioning variable provided to the flow model. Could you elaborate on how this leads to significantly improved results?

9. Figure 5(a) is not very informative, as it is already well-known that DDPMs are slow at inference. Did you try DDIM or other acceleration techniques?

10. Figure 6 is small and the text is difficult to read. Moreover, it is hard to visually inspect generated images under different conditions. You could complement the visual results with additional empirical metrics, as done in [1].

References:

[1] PhenDiff: Revealing Subtle Phenotypes with Diffusion Models in Real Images, Bourou et al.

[2] Revealing invisible cell phenotypes with conditional generative modeling, Lamiable et al.

[3] Flow Straight and Fast: Learning to Generate and Transfer Data with Rectified Flow, Liu et al.

[4] Denoising Diffusion Implicit Models, Song et al.

---

> ### Author Response · Authors · 2025-11-24
> **We have addressed each weakness point by point.**
>
> **Dear reviewer, thank you for your valid and fair feedback. We have now addressed each point. We believe our updated manuscript has substantial technical and applied contributions to the field of multi-modal biological perturbation modelling, and request a reassessment of our work. To the best of our knowledge, ours is the first work to tackle this problem. We are committed to improving the paper further for the final version, and are willing to have a discussion and address all other questions.**
>
> **W1:** We have moved the extensive architecture details to the appendix. The dataset curation and preprocessing from many scattered datapoints from Ginkgo Bioworks GDPx is a major contribution of our work. We will be releasing the full processed dataset for the scientific community to test our models and train new ones. We have also provided the gene enrichment analysis results in lines 509-525.
>
> **W2:** We have fixed the small text in all figures. Please let us know if it is clear now. We are committed to enhancing the presentation and clarity of the paper however the reviewer wishes.
>
> **W3:** We have added the following sentence: "For instance, recent efforts focus purely on image-to-image translation, such as PhenDiff (Bourou et al., 2023), which uses a conditional diffusion model, and Lamiable et al. (2023), which employs conditional GANs. Critically, these methods generate a cell image in one condition given an image from another but operate without any transcriptomic context."
>
> **W4:** We have now added baselines for single modality in Tables 1 and 2 at the top. The MLP baseline in Table 1 is a simple linear fully connected model with MSE and PC loss. The VAE PRNet [1] baseline is a popular method for perturbation guided transcriptome prediction. Our PertFlow method beats both baselines. Similarly the UNet baseline is a deterministic UNet autoencoder model to generate cellular images with an MSE loss. The UNet PhenDiff [2] model is a direct adaptation of the cited method by the reviewer with our custom drug encoder. Our PertFlow model again beats both baselines.
>
> [1] Qi, X., Zhao, L., Tian, C. et al. Predicting transcriptional responses to novel chemical perturbations using deep generative model for drug discovery. Nat Commun 15, 9256 (2024).
>
> [2] PhenDiff: Revealing Subtle Phenotypes with Diffusion Models in Real Images, Bourou et al.
>
> **W5:** Thank you for this suggestion. We have moved the technical architecture part in the appendix and have also numbered the equations for clarity. We have provided a more cleaner and concise summary of the architecture in lines 202-245.
>
> **W6:** We have further updated Figure 4 in view of new results from PhenDiff. We have also updated the text explaining the new results in lines 313-369 and 383-412.
>
> We have split the UMAP and t-SNE results into Figures 8, 9, 10. The key takeaway is that the cross-modal attention mechanism successfully establishes correspondence mappings between RNA and image modalities while preserving their distinct representational structures. Quantitative analysis reveals that pairwise alignment distances between corresponding RNA-image samples average 16.48 (UMAP), and 37.44 (t-SNE) in the joint embedding space, with UMAP achieving the tightest correspondence.
>
> **W7:** Our contribution is not an incremental upgrade of simply beating single modality methods by a few numbers on an existing single modality benchmark by adding extra encoders.
>
> We have created a new benchmark by curating data from multiple Ginkgo Bioworks datasets and a novel methodology for jointly modelling transcriptomics and images with drug perturbation conditioning. The core contribution and novelty is modelling a joint space that aids in generation of cell morphology and predicting gene expression data. Most methods have not even tried joint modelling since there was no curated data available. Our contributions are:
>
> 1. First method to jointly predict transcriptomic and generate morphological responses to chemical perturbations
> 2. A shared embedding space integrating control RNA-seq, control images, and drug metadata to model complex dependencies
> 3. Multi-token cross-attention to align molecular and morphological features across modalities
>
> We established the benchmark for state-of-the-art performance on the GDPx3 dataset, improving cross-modal alignment and prediction quality over single-modality and diffusion baselines. PertFlow could support downstream applications in virtual drug screening, mechanism discovery, and integrated pharmacological modeling by enabling joint prediction of RNA-seq and image responses to perturbations; a capability, to our knowledge, the first and unique among current methods.
>
> In view of these major contributions, we request the reviewer to reassess the novelty of our work.

---

> ### Author Response · Authors · 2025-11-24
> **We have addressed each question point by point.**
>
> **Q1:** We added loss weight ratio experiments in Table 3. The 0.5-0.5 ratio achieves the best tradeoff: higher RNA weight improves RNA accuracy but worsens image quality (higher FID), while higher image weight improves SSIM/PSNR but reduces RNA correlation.
>
> **Q2:** The shared representation h_shared conditions both heads. For RNA-seq: x^treat_rna = TranscriptomeHead(h_shared) via MLP projection. For images: h_shared is projected to c_img = ImageUNet(h_shared), then fed into UNet cross-attention blocks at layers 2-5. At each timestep t, the UNet computes v_θ(x_t, t, c_img). The novelty is the multi-modal conditioning signal integrating RNA-seq, imaging, and drug metadata.
>
> **Q3:** The comparison should be "does joint modeling enable capabilities single-modality approaches cannot achieve" rather than "can joint models beat single-modality models." PertFlow achieves competitive RNA-seq performance while simultaneously generating morphological images - a capability PertRNA lacks. PertFlow outperforms PertImage by leveraging transcriptomic information. The value is enabling new applications: virtual screening evaluating molecular and phenotypic effects, mechanism discovery linking gene expression to morphology, and integrated pharmacological modeling capturing multi-scale responses.
>
> **Q4:** We included PhenDiff as baseline in Table 2 and Figure 4. PhenDiff achieves SSIM 0.189, PSNR 9.64, LPIPS 0.613, FID 62.50 - PertFlow outperforms across all metrics. Expert pathologist evaluation: PertFlow scored 7.11/7.89 vs PhenDiff 6.11/6.33.
>
> **Q5:** All metrics computed on test set (3,448 samples, 20% of data, stratified by compound-cell line). Evaluation is per treatment pair: each control-drug-condition generates treatment image compared to ground truth. SSIM/PSNR/LPIPS computed per-sample and averaged; FID computed globally. We evaluate treatment-specific responses, not unconditional generation.
>
> **Q6:** Font sizes increased in all figures. Consistent sizing across tables and figures. Tables kept compact to avoid full-page width. All content is high DPI for zooming.
>
> **Q7:** Three PertDiff variants differ in training objective:
> - **PertDiff_N**: noise prediction (ε from x_t), FID 246.01 - poor due to signal-noise disentanglement at low SNR
> - **PertDiff_x0**: direct clean target prediction, FID 73.63 - better stability but lacks theoretical grounding
> - **PertDiff_V**: velocity prediction (v = α_t ε - σ_t x_0), FID 55.92 - best diffusion performance
>
> Our rectified flow (FID 24.06) outperforms PertDiff_V via straight-line trajectories vs noisy curved paths, enabling faster sampling (7-10 vs 50+ steps) and more stable training.
>
> **Q8:** Improved performance comes from the entire multi-modal framework, not just conditioning variable. PertImage conditions only on drug metadata and control image. PertFlow conditions on h_shared integrating control RNA-seq, control image, and drug information via cross-modal attention. This learns molecular-morphological correspondences: camptothecin activating DNA damage pathways associates with nuclear fragmentation; dabrafenib modulating MAPK signaling predicts cytoplasmic reorganization. Ablations show: removing knowledge graph (-KG) degrades performance; removing triplet loss worsens image quality (FID 106.72 vs 24.06). The conditioning captures learned multi-modal representations of biologically meaningful relationships.
>
> **Q9:** Results are from DDIM not DDPM. DDPM requires 100+ steps, DDIM requires 20-50 steps. We test multiple diffusion techniques with quantitative results but show only DDIM in NFE comparison.
>
> **Q10:** Enlarged Figure 6, split into Figures 6-7, increased text size. Table 2 provides comprehensive quantitative evaluation: SSIM (structural), PSNR (signal fidelity), LPIPS (perceptual), FID (distributional), plus expert pathologist evaluation (Figure 4). Generated images correctly capture drug-specific morphological signatures confirmed by computational metrics and expert evaluation.
>
> If the reviewer has specific metrics in mind beyond these standard benchmarks, we would be happy to include them. We also emphasize that the key validation is biological: the generated images correctly capture drug-specific morphological signatures as confirmed by both computational metrics and expert evaluation. Visual inspection serves to qualitatively demonstrate these biological effects, while quantitative metrics provide rigorous performance assessment.

---

### Author Response · Authors · 2025-11-24
**Comment for all reviewers.**

**Dear reviewers, we would like to emphasize that no existing method jointly generates both transcriptomic and morphological responses to chemical perturbations. Our contribution is not an incremental upgrade of simply beating single modality methods by few numbers on an existing single modality benchmark, rather, it represents a significant advance over current methodologies. It is essential that the reviewers understand the scope of the problem and kindly request them to re-evaluate the paper.**

- We have created a new multi-modal benchmark by curating data from multiple Ginkgo Bioworks datasets and a novel methodology for jointly modelling transcriptomics and images with drug perturbation conditioning.

- This direction of multi-modal research will immensely help in advancing AI for science and is an invaluable contribution to the applied ML for biology community.

- We will release the processed dataset, with the full training code and inference and evaluation scripts.

- The core contribution and novelty is in modelling a joint space that aids in generation of cell morphology and predicting gene expression data. To the best of our knowledge, ours is the first work to tackle this problem.

**We have now address all the concerns raised by reviewers point by point and significantly improved the paper based on their suggestions.**

- New baseline results have been reported on single modality methods in Tables 1 and 2.

- Metrics reported on sensitivity analysis for loss weights in Tables 3 and 4.

- New and improved images with extremely high DPI and larger text size.

- Further UMAP and t-sne results to visualize latent space with metrics.

- Cleaned up equation heavy architecture details and added equation numbers.

- Pathologists have given descriptions on cellular morphology of both generated and real treatment images.

- Gene-enrichment analysis moved to the main text.

- We will move extra details to the appendix for the final version to fit the page limit.

**We are open to having a discussion with all the reviewers and performing further experiments, and answering any questions.**

---

### Note · Authors · 2026-01-26

I have read and agree with the venue's withdrawal policy on behalf of myself and my co-authors.

---

### Meta-Review · Area_Chair_2iLB · 2025-12-19

**Summary:**

This paper looks at the problem of generating morphology images alongside gene expression profiles in response to perturbations.

The main criticism, shared by all reviewers, is that the authors essentially create a new data set for this problem and show results only there, without comparing to unimodal benchmarks. The authors rebutted that indeed, this data set is the first of its kind and that the goal is not to outperform unimodal models. Yet, I would agree with the reviewers that more thorough evaluation is needed, alongside stronger baselines. The data set has clear value, although is essentially a careful combination of existing datasets. One clear negative to me is that the authors did not submit the dataset alongside the paper, so it was impossible for the reviewers to inspect it. I don’t think it makes a strong contribution this way.

I am not sure whether the model is particularly interesting; the fact that multiple modalities can be generated together is hardly novel in ML. This is not necessarily a minus on its own, but then I would question whether this paper would be at home in a biology journal. If the authors had shown that the joint modeling improves over unimodal models that would have been quite interesting in my opinion, yet the authors maintained that it would not be in the scope of their submission.

Overall, I think this paper needs at best a new round of reviews, at worst, completely new experiments. Therefore, I recommend rejection at this point.

**Reviewer Concerns:**

The reviewers raised a large number of concerns, but unanimously agree that the paper has core limitations in its positioning with respect to baselines and related works. These were not answered satisfactorily in the rebuttal.

**Reviewer Scores:**

It's possible that all reviewers may have marginally increased their scores, but given the initial starting point and the fact that the authors disagreed with the reviewers on their concerns, I am certain the paper would have been rejected even with full participation in the discussion.

---

### Decision · Program_Chairs · 2026-01-26

Reject